**Nonlinear turnover rates of soil carbon following cultivation of native grasslands and**

**subsequent afforestation of croplands**

Guillermo Hernandez-Ramirez*
Department of Renewable Resources
University of Alberta
Edmonton, AB, T6G2R3
Canada
*Corresponding author. Tel.: +1 780 492 2428, E-mail address: ghernand@ualberta.ca.

Thomas J. Sauer
USDA-ARS,
National Laboratory for Agriculture and Environment,
Ames, IA 50011,
USA

Yury G. Chendev
Department of Natural Resources Management and Land Cadastre,
Belgorod State University,
85 Pobeda Street, Belgorod 308015,
Russia

Alexander N. Gennadiev
Lomonosov Moscow State University,
Faculty of Geography,
119991, Moscow, GSP-1, 1 Leninskiye Gory,
Russia

**Abstract**

Land use conversions can strongly impact soil organic matter (SOM) storage, which

creates paramount opportunities for sequestering atmospheric carbon into the soil. It is known
that land uses such as annual cropping and afforestation can decrease and increase SOM,
respectively; however, the rates of these changes over time remain elusive. This study focused on
extracting the kinetics (k) of turnover rates that describe these long-term changes in soil C
storage and also quantifying the sources of soil C. We used topsoil organic carbon density and
$\delta^{13}C$ isotopic composition data from multiple chronosequences and paired sites in Russia and
United States. Reconstruction of soil C storage trajectory over 250 years following conversion
from native grassland to continual annual cropland revealed a C depletion rate of 0.010 years$^{-1}$
(first-order k rate constant), which translates into a mean residence time (MRT) of 100 years
($R^2 \geq 0.90$). Conversely, soil C accretion was observed over 70 years following afforestation of
annual croplands at a much faster k rate of 0.055 years$^{-1}$. The corresponding MRT was only 18
years ($R^2 = 0.997$) after a lag phase of 5 years. Over these 23 years of afforestation, trees
contributed 14 Mg C Ha$^{-1}$ to soil C accrual in the 0 to 15 cm depth increment. This tree-C
contribution reached 22 Mg C Ha$^{-1}$ at 70 years after tree planting. Over these 70 years of
afforestation, the proportion of tree-C to whole soil C increased to reach a sizeable 79%.
Furthermore, assuming steady state of soil C in the adjacent croplands, we also estimated that
45% of the prairie-C existent at time of tree planting was still present in the afforested soils 70
years later. As intrinsic of k modelling, the derived turnover rates that represent soil C changes
over time are nonlinear. Soil C changes were much more dynamic during the first decades
following a land use conversion than afterwards when the new land use system approached
equilibrium. Collectively, results substantiated that C sequestration in afforested lands is a
suitable means to proactively mitigate escalating climate change within a typical person's
lifetime, as indicated by MRTs of few decades.



## 1 Introduction

The global effects of escalating climate change are in part driven by land use choices. Indeed, implementing certain land use changes can worsen or in other cases mitigate emissions of greenhouse gases from terrestrial ecosystems to the atmosphere (Paustian et al., 1992; Post and Kwon, 2000; Thilakarathna and Hernandez-Ramirez, 2021). Essentially, land use options that unintentionally accelerate biological oxidation of soil organic matter (SOM) contribute over time to atmospheric carbon dioxide concentrations (Sauer et al., 2007; Laganiere et al., 2010; Li et al., 2018), providing a portion of the radiative forcing that has been causing part of the global warming effect over the last decades (Guo and Gifford, 2002; Parry et al. 2007). Relative to contrasting types of land use systems, annual croplands commonly showed SOM depletion and marked reductions in soil C storage, in particular compared with their natural ecosystem counterparts (Chendev et al., 2015b; Hebb et al., 2017; Kiani et al., 2017).

Contrary to the potentially detrimental effects of annual cropping on SOM accumulation and overall soil quality (Guenette and Hernandez-Ramirez, 2018; Laganiere et al., 2010; Kiani et al., 2020), tree planting offer multiple environmental services and societal benefits (Hernandez-Ramirez et al., 2012; Sauer et al., 2012; Zhang et al., 2020). For instance, removing C from the atmosphere is a paramount contribution by trees (Guo and Gifford, 2002; Li et al., 2012; Li et al., 2018). In effect, soil C accrual (Paul et al., 2002; Dhillon and Van Rees, 2017; Khaleel et al., 2020) and stabilization (Hernandez-Ramirez et al., 2011; Wang et al., 2016; Quesada et al., 2020) beneath trees have been recognized as an effective means for sequestering atmospheric C. In addition to soil accruals, diverse microbial communities can flourish beneath mature trees (Kiani et al., 2017). Additional functions by tree vegetation include improving air quality, enhanced microclimate, and erosion control (Sauer et al., 2007; Hernandez-Ramirez et al., 2012;

Chendev et al., 2015b). Establishing agroforestry practices such as shelterbelt systems within
annual croplands can provide a balance between continual food production and tree benefits with
only a fraction of the landscape occupied by trees (Amadi et al., 2016; Dhillon and Van Rees,

2017).

Sequestering C in soils is governed by the balance of inputs of plant C with

decomposition and stabilization processes (Hernandez-Ramirez et al., 2009; Kiani et al., 2017; Li
et al., 2018). This overall functioning of C-related biology and cycling in soils can be described
as the turnover of soil C. Collectively, SOM mineralization, gains and losses, and net accrual can
be numerically integrated into C turnover rates (Richter et al., 1999; Hernandez-Ramirez et al.,
2011; Xiong et al., 2020). For instance, dynamic rates of net depletion of SOM pools caused by
continual cropping or tree contributions to soil C accretion and cycling under afforestation can
both be captured as C turnover rates (Guo and Gifford, 2002; Hu et al. 2013). Nevertheless, since
using linear rates to describe changes in soil C often leads to poor estimates of C inventories and
sequestration, soil C accrual rates need to be derived as nonlinear rates to accurately predict the
trajectory of soil C changes with time following land use conversions (Post and Kwon, 2000;
Garten 2002). Moreover, the direction and net rates of SOM accrual as a response to land use
changes need to be assessed in the long term (i.e., ranging from decadal to centurial scales)
(Paustian et al., 1992; Hernandez-Ramirez et al., 2011). This new knowledge will inform how
lasting these effects of land management options on soil C storage are, enabling predictions of
future soil C sequestration (Richter et al., 1999; Guo and Gifford, 2002). Our study endeavors to
address and fill these knowledge gaps.

Testing accrual rates of SOM is still lacking in the literature. Previous studies have
evaluated soil C turnover rates as a function of changes in land management only over one or a
few decades (Jastrow et al., 1996; Hernandez-Ramirez et al., 2011; Mary et al., 2020); however,
the underlying assumption of asymptotic behavior in the rate of soil C change has rarely been
verified over longer periods such as over centuries. Likewise, earlier studies examining dynamics
of soil carbon in continuous annual croplands have suggested typical MRTs of 117 yr (Huggins
et al., 1998) and 57 yr (Collins et al., 1999); however, it is still unclear how long-term land use
changes from native grasslands to annual croplands and from annual croplands into afforestation
can impact the turnover rates of soil C over centuries.
In this study, we compiled soil C storage data from several field sites comparing land use
systems in Russia and United States (Chendev et al., 2015a, 2015b) in conjunction with
published (Hernandez-Ramirez et al., 2011) and newly-available soil $^{13}$C isotope data. Based on
these data assemblage, we now focused on evaluating the long-term turnover rates of soil C as a
function of land use changes from native grasslands to annual croplands and subsequent
converting annual croplands into afforestation. We aimed at extracting turnover rates of soil C
depletion or accretion, which can enable future predictions of soil C storage depending on land
use systems. Also, our study quantified and documented the contributions of tree biomass-C to
soil C that was newly-accrued following afforestation. We further examined the stage and net
losses of the C that existent in the soil under annual croplands prior to tree planting.

**2 Materials and Methods**
**2.1 Chronosequences from native grasslands to annual croplands in Russia**
Three long-term chronosequences of land use conversion (i.e., a range of different
duration of cultivation) were used for extracting turnover parameters. These land use
chronosequences were situated in Belgorod oblast, Russia within the districts of Prokhorovskiy
(50°57′ N, 36°44′ E), Gubkinskiy (51°03′ N, 37°22′ E) and Ivnyanskiy (51°06′ N, 36°24′ E) as
previously described by Chendev et al. (2015a) (Fig. 1). Following chronosequence methods as
described by Laganiere et al. (2010), each chronosequence had four or five age-sites
encompassing a native grassland site that represented the time zero of conversion from grassland
to annual cropland. These native grasslands were undisturbed steppe dominated by plant species
with C3 photosynthetic pathway. The ages of the cropland sites were established through
historical records and geographic approaches described by Chendev et al. (2012, 2015b).
Additional information about the study sites is available at Chendev et al. (2015a), while the
focus in our study remains on developing models and extracting parameters of C turnover rates.
Typical crops species included cereals, sunflower (*Helianthus annuus*) and beet (*Beta vulgaris*)
managed under conventional tillage operations. Within the study region, soils were classified as
loamy Chernozems (Russian Soil Classification System), annual precipitation ranged between
480 and 580 mm and air temperature between 5.3 and 5.8 °C (Chendev et al., 2015a).
Composite soil samples (3 subsamples per sample) with at least 12 sampling locations
per age-site were collected using the core method in 10 cm depth increments to 1 meter depth.
Field moist soil samples were passed through 8- and 2-mm sieves, air dried, and ground with a
roller mill (Bailey Manufacturing Inc., Norwalk, IA) to create a fine powder consistency.
Identifiable plant materials were removed prior to grinding (Hernandez-Ramirez et al., 2011;
Chendev et al., 2015b).
Soil organic C mass density for the 0 to 30 cm depth was calculated as the sum of
products of organic C concentration (Hernandez-Ramirez et al., 2009), bulk density and soil
layer thickness, with units of Mg C ha$^{-1}$.
First-order kinetic modelling follows:
$C_{(t)} = C_e + (C_o - C_e) \, e^{-kt}$                                                    [1]
where $C_e$ is soil C storage at the oldest cropland site within each chronosequence which was
assumed to be at new dynamic equilibrium (i.e., C inputs = C outputs), $C_o$ is soil C storage at the
native grassland site which was assumed to be the initial time of land use conversion from native
grassland to annual croplands (time zero), k is the fitted first-order kinetic rate constant (yr$^{-1}$),
which is equivalent to C turnover rate or net C mineralization (in the case of net C decreases),
and t stands for time (yr). In the case of increases in soil C over time, turnover rates become
equivalent to accretion rates. It is possible to model the soil C storage for each year, and hence,
the difference between consecutive years provides an estimation of the annual net C change (Mg
C ha$^{-1}$ yr$^{-1}$). First-order kinetic modelling (Eq. [1]) assumes: (i) balance between C inputs and C
outputs and (ii) steady state conditions (i.e., $\delta C/\delta t = 0$) (Jastrow et al., 1996; Follett et al., 1997;
Hernandez-Ramirez et al., 2011). At the various study sites, the terrain slopes ranged up to 2%,
with the exception of the Huron site that had 3% slope. Hence, the general topography in our
study sites was classified as flat. We assume that semiarid climate, enough vegetation cover and
low slope limit water and wind erosion. Given the dominant flat topography and low rainfall
amounts, we also assumed negligible C removals or additions due to erosion or deposition.
Mean residence time (MRT) of organic C in the soil was calculated as reciprocal of k.
Concomitantly, half-life of organic C in the soil was calculated as follows:
Half-life = ln(2)/k                                                    [2]
Note that under equilibrium, C output$_e$ is also equivalent to C input$_e$, and they correspond to the
annual C that enters and exits the soil C pool, respectively.

The performance of the derived first-order kinetic modelling was evaluated with the

normalized root mean square error (RMSEn) (Guenette and Hernandez-Ramirez, 2018; Kiani et
al., 2020), coefficient of determination ($R^2$) as well as a leave-one-out cross-validation of
predicted versus measured C (n= 6). Within the cross-validation, we tested the regression
coefficient ($\beta_1$) of a linear regression established for predicted vs. measured C against the 1:1
line.

**2.2 Comparison of adjacent paired sites in Russia: native grasslands, annual croplands and**

**shelterbelts (trees)**

In addition to the 3 abovementioned chronosequences, 3 additional sites were studied in

Russia: Streletskaya Steppe situated within Kursk oblast (51°32′ N, 36°05′ E), Yamskaya Steppe
in Belgorod oblast (51°11′ N, 37°37′ E), and Kamennaya Steppe in Voronezh oblast (51°02′ N,
40°44′ E) (Fig. 1). Soils at all these paired sites were classified as loamy Chernozems. Following
field methods as described by Laganiere et al. (2010), each site encompassed adjacent locations
representing three land uses: native grassland, annual croplands and broadleaf shelterbelts, as
described by Chendev et al. (2015b). Soil sample collections were conducted similar as
described above. Briefly, composite soil samples (3 subsamples per sample) were collected from
the native grassland (n= 6 composite soil samples), annual croplands (24) and shelterbelts (18) in
summer 2012. At time of soil sample collection, the ages of the annual croplands in Streletskaya
and Yamskaya were at least 140 years, and at least 145 years of age in Kamennaya. In all 3 sites,
the shelterbelts had been planted 55 years prior to soil sample collection. Tree species in the
shelterbelts include silver birch (*Betula verrucosa*), Manitoba maple (*Acer negundo*), and
English oak (*Quercus robur*). Long-term mean annual precipitations at Streletskaya, Yamskaya
and Kamennaya correspond to 580, 530 and 480 mm yr$^{-1}$, respectively.

It is noted that although trees at the Streletskaya, Yamskaya and Kamennaya sites were

planted 55 years prior to soil sample collection, for first-order kinetics modelling purposes, the
tree-C contributions to soil C accrual were accounted beginning from 50 years prior to soil
sample collection. This assumption is based on a literature review by Paul et al. (2002) who
suggested a lag phase of 5 years for tree-C contributions to effective start contributing to net
storage of soil C. Moreover, because of the uncertainty of how close the afforested soils were to
steady state and equilibrium of soil C storage, we evaluated two scenarios of first-order kinetics
modelling using Eq. [1]. We assessed trajectory 'A' under the premise that full steady state has
been reached at time of soil sample collection, and also trajectory 'B' where we assumed that the
C storage in these afforested soils had asymptotically reached 95% of the theoretical equilibrium
or ceiling capacity. We reported both trajectories and their associated C accretion rates (k).

The rationale for implementing the 95% scenario (trajectory 'B') arises from the

uncertainty of whether full C equilibrium has been reached. As the final phases in an asymptotic
trajectory are incremental, we undertook the evaluation of a narrow but identifiable deviation
below full C equilibrium. The choice of 95% represents that an analytical precision for
quantifying soil carbon can typically be found within 5%. In other words, based on a principle of
detection limit in soil C measurements, a 95% can be considered a minimal deviation from full
equilibrium (100%) that is already discernable, but still related to the general variability of the
quantification method and associated results. Hence, we took the freedom to assess this plausible
range at and below full theoretical equilibrium, with the soil C storage having reached 100% or
95% of the ceiling capacity, respectively. From a broader perspective, this alternative 95%
scenario also explores and represents the prospect that the soils beneath the shelterbelt could still
be incrementally accruing C even 55 years after tree planting.

**2.3 Pairwise comparisons in United States: annual croplands and afforestation**

Three field sites were studied within the Northern Great Plains of United States near the
cities of Huron (South Dakota, 44°15′ N, 98° 15′ W), Norfolk (Nebraska, 42° 03′ N, 97° 22′ W)
and Mead (Nebraska, 41°9´ N, 96°29´ W) (Fig. 1). While the focus remained on investigating the
turnover rates of soil C as a function of land use changes, it is noted that pedogenic ages of the
soils sampled in United States were relatively shorter than the soils studied in Russian sites. This
is because of the differences in geological times of parent materials exposure on the ground
surface between geographic regions since the sites in United States experienced the last
glaciation (i.e., Wisconsin glaciation).
The 3 sites in United States encompassed afforested areas and adjacent annual croplands,
co-located in paired sites as Laganiere et al. (2010). The native vegetation at the sites had been
tallgrass prairie (e.g., big bluestem *Andropogon gerardii* Vitman), which had been converted into
annual croplands, and trees were subsequently planted in areas of the croplands. Afforestation
took the forms of shelterbelts in Norfolk and Mead, and a forest plantation in Huron. The Huron
site also had an adjacent field with a representative undisturbed native prairie, which was also
sampled as a reference. In contrast to the long-term croplands in Norfolk and Mead, the cropland
at Huron had only 21 years since conversion from native grassland at time of soil sample
collection. Field sample collections were conducted in 2004 in Mead, and in 2012 in both
Norfolk and Huron. The trees had been planted 19, 35 and 70 years prior to soil sample
collections in Huron, Mead and Norfolk, respectively. Tree species included green ash (*Fraxinus*
*pennsylvanica* Marshall), red cedar (*Juniperus virginiana* L.), and oak (*Quercus macrocarpa*) in
Huron; red cedar, scotch pine (*Pinus sylvestris* L.), and cottonwood (*Populus deltoides* Bartram)
in Mead; Siberian elm (*Ulmus pumila*), red mulberry (*Morus rubra*), and cottonwood in Norfolk.
Annual croplands were managed under conventional farming practices. Annual crop species at
the study sites included wheat (*Triticum aestivum* L.), corn (Zea mays L.), soybean [Glycine max
(L.) Merr.], and sorghum (*Sorghum bicolor* L. Moench). An alfalfa (*Medicago sativa* L.) forage
field adjacent to the shelterbelt in Norfolk was also sampled. Long-term mean annual
precipitation in Huron, Norfolk and Mead were 582, 696 and 747 mm yr$^{-1}$, respectively. Long-
term mean annual air temperature in Huron, Norfolk and Mead were 7.7, 9.6, 9.9 °C,
respectively. Overall, study sites had soil pH near neutral and textures between loamy sand to
silty clay loam (Table 1)

Field methods of soil sample collections had previously been described in related reports

by Chendev et al. (2015a) for the Huron and Norfolk sites as well as by Sauer et al. (2007) for
Mead. Briefly, spatial grid patterns were established with composite samples (n= 4) collected
from each sampling location. Total grid sampling locations were 118 at Mead, 48 at Huron, and
42 at Norfolk. Plant tissue samples of the dominant species were also collected from each study
site.

Organic C concentration and $\delta^{13}$C isotopic composition were determined in all soil and

plant samples via the dry combustion method using a Fison NA 15000 Elemental Analyzer
(ThermoQuest Corp., Austin, TX) interfaced to an isotope-ratio mass spectrometer Delta V
Advantage (Thermo Fisher Scientific, Waltham, MA). Pee Dee Belemnite was used as standard
and analytical precision of $\delta^{13}$C measurements was 0.06‰. The $\delta^{13}$C isotopic ratio was
expressed as:
$\delta^{13}$C (‰) = [($^{13}$C/$^{12}$C sample) / ($^{13}$C/$^{12}$C standard) - 1] $\times$ 1000          [3]
When integrating multiple soil layers of a profile, averages of $\delta^{13}C$ were weighted by the
soil C mass density at the corresponding soil layers.
Prior to land use conversion to croplands, the native grasslands in United States were
undisturbed and dominated by plant species with C4 photosynthetic pathway, with certain mixed
presence of C3 species. Based on this legacy contribution of prairie vegetation to soil C over the
Holocene, approaches based on stable isotope signatures became feasible in the 3 sites in United
States. Furthermore, the ability to use a C stable isotope approach to partition the current soil C
into two specific C pools (i.e., remaining prairie-C and new tree-C) requires a constraining
assumption that much of the plant residues added yearly over the annual cropping stages
decomposes during the following growing season (Gregorich et al., 2017). Therefore, this
premise entails that most of C in crop residues enters the soil to become lost back to the
atmosphere within a year, and hence, having near-negligible contributions to changes in both net
C accrual and $\delta^{13}C$ isotopic composition in the soil. This method enabled us to examine the
sources of soil C and also derive the turnover rates of these soil C sources. This primarily applies
because trees are C3 species. This approach assumed that the differences in $^{13}C$ isotopic
signatures between C4-C3 mixed (native grassland) and C3 (trees) overrides any potential
differential effect of C isotopic fractionations during SOM decomposition of C3 vs. C4
substrates, between aboveground and below ground plant materials (roots vs. litter), or because
of SOM interactions with soil mineral surfaces (Martin et al., 1990; Hernandez-Ramirez et al.,
2011). Assuming mass conservation, the measured soil C storages were allocated into two
sources: (i) new tree-C and (ii) remaining prairie-C (native soil) as follows:
Tree-C + Prairie-C = 1                                                                      [4]
Tree-C = ($\delta^{13}C$ afforested soil - $\delta^{13}C$ native soil)/($\delta^{13}C$ tree - $\delta^{13}C$ native soil)     [5]
It was inferred that all soil C different from the C identified as new 'Tree-C' was
preexisting soil C attributable to remaining 'Prairie-C'. Likewise, we assumed that the '$\delta^{13}$C
native soil' were represented reasonably well by the $\delta^{13}$C measured in soil samples taken from
the annually-cropped fields adjacent to the afforested soils. Although they were adjacent, the
sampling locations providing the '$\delta^{13}$C native soil' were sufficiently distant from afforested areas
to preclude influence of trees on soil $\delta^{13}$C. The '$\delta^{13}$C native soil' were -17.3‰ in Huron, -17.0‰
in Mead, and -17.5‰ in Norfolk, which are noted to be consistent with each other as these sites
share a common natural history of tallgrass prairie native vegetation. These '$\delta^{13}$C native soil' are
also consistent with earlier measurements in prairie soils by Follett et al. (1997) and Hernandez-
Ramirez et al. (2011). Furthermore, the $\delta^{13}$C measured in tree samples averaged -27.6‰ in
Huron, -26.6‰ in Mead, and -27.9‰ in Norfolk as typical isotopic compositions of C3 plant
species. At the Norfolk site, tissue samples of alfalfa canopy yielded -27.4‰. Also, in Norfolk,
'$\delta^{13}$C afforested soil' included all data from the soil samples taken within 10 m north and 10 m
south from the center of the shelterbelt. In Huron, '$\delta^{13}$C afforested soil' sample included all data
from soil samples taken at least 17 m away from the edge between the forest plantation and the
adjacent annually-cropped field. In Mead, as earlier presented by Hernandez-Ramirez et al.
(2011), '$\delta^{13}$C afforested soil' sample included all data from soil samples collected between the
existing two tree rows.
The mass densities of soil C derived from new tree-C and remaining prairie-C were
calculated by multiplying whole soil C storage beneath the trees with the corresponding fractions
expressed in Eq. [4].
When assessing first-order kinetics modelling (Eq. [1]) of soil C beneath trees, we
assumed that the remaining prairie-C in afforested soils at time of soil sample collection was at
steady state and had also reached new equilibria in the case of the long-term annual croplands at
Norfolk and Mead. In the specific case of Huron, because the afforested soil had experienced
only two years of annual cropping prior to tree planting, we assumed that the native grassland
prior to land use conversion to annual cropping was at equilibrium and steady state.

As abovementioned, the shift from C4-dominated to C3 vegetation in the case of the

afforested soils in United States enabled us to methodologically apportion the sources of soil C
and to identify these direct contributions from trees to increasing soil C storage. In the case of the
Russian sites, soil $^{13}$C isotope composition does not resolve these C sources because of the lack
of shift between C4 and C3 vegetation in the natural history of these landscapes.

Relationships between allocations of soil C sources (% tree-C, and % prairie-C) and time

since tree planting (years) were examined through linear regression analyses. Likewise, a similar
linear regression was developed for the proportions of remaining prairie-C in the afforested soils
relative to whole C present in the adjacent cropland soils. We used SigmaStat Version 4.0
software (Systat Software, San Jose, CA) and an α critical level of 0.05. Where error terms (±)
are presented, they correspond to the standard errors of the means.

**3 Results**
**3.1 Soil C after conversion of grasslands to cropland and then shelterbelt: Russian cases**

Long-term cultivation of native grasslands decreased soil organic C storage in a nonlinear

fashion (Fig. 2). Within each of the 3 available land use chronosequences (i.e., each
encompassing a range of different duration of cropping since land use conversion), the declining
trajectory of soil C was represented reasonably well by first-order kinetic modelling. The
RMSEn were all lower than 4% and the $R^2$ were greater than 90%, which supports the suitable
performance of k models (Fig. 2A-C). Likewise, cross-validation results of cropland soils within
the age range from 10 to 200 years further indicated the high accuracy of k predictions when
compared with the 1:1 agreement line. This was based on a non-significant t-test with $\beta_1=1$ as
null hypothesis (Fig. 2D).
Within the soil layer of 0 to 30 cm depth in the chronosequences in Belgorod, Russia,
turnover rates (k) of soil C ranged from 0.0091 to 0.0183 $yr^{-1}$ in Gubkinskiy and Prokhorovskiy,
respectively. Over the entire time spans of the 3 chronosequences (up to 250 years old), net soil
C losses were in the relatively narrow range from 31.2 Mg C $ha^{-1}$ in Prokhorovskiy (Fig. 2A) to
36.9 Mg C $ha^{-1}$ in Gubkinskiy (Fig. 2B). Focusing on these losses of the preexisting soil C, the
estimated lapses for half of these C losses to take place was between 38 and 76 years after the
time of land use conversion from native grassland to annual cropland (Fig. 2A and Fig. 2B,
respectively). It is noted that new dynamic equilibria were assumed to have taken place in the
oldest cropland soil within each chronosequence (i.e., > 200 yrs) as part of first-order kinetics
modelling. Furthermore, when examining the assumption of steady state (i.e., $\delta C/\delta t = 0$), soil C
trajectories at both Prokhorovskiy and Ivnyanskiy sites showed reasonable approximations to
this premise, with relatively low annual C losses occurring towards the end of these
chronosequences. On the other hand, the Gubkinskiy site still exhibited vigorous C losses at the
end of this chronosequence, challenging the steady state assumption at this site. In further details,
the trajectory at the Prokhorovskiy site showed estimated C losses of only 9.4 Kg C $ha^{-1}$ $yr^{-1}$
during the last year of this chronosequence (Fig. 2A), which can be considered negligible and in
clear agreement with a steady state condition. Conversely, the last year of the Gubkinskiy
chronosequence still showed a C loss of 34.5 Kg C $ha^{-1}$ $yr^{-1}$. It is noteworthy that these annual C
outputs typically take the form of soil respiratory losses ($CO_2$-C) that are resultant from
microbial mineralization of existing SOM. These estimations of net changes at near steady state
do not account for the $CO_2$ derived from the decomposition of recently-added plant residues, but
just the net change in SOM-C (Fig. 2A). For comparison purposes, the k-modelled trajectory of
the Prokhorovskiy chronosequence had estimated C losses of 566 Kg C ha$^{-1}$ yr$^{-1}$ during the very
first year after conversion from native grassland to annual cropland.

Based on the assessed pairwise comparisons, afforestation in the form of shelterbelts

replenished soil C storage after long-term annual cropping had led to decreasing soil C compared
with adjacent native grasslands (Table 2). Of the substantial soil C storage that had been depleted
over time during annual cropping (i.e., -18.9 ± 5.3 Mg C ha$^{-1}$), afforestation replenished on
average 81% of these cropping-induced C losses (Table 2).

Using the insights gained from both chronosequences (Fig. 2) and pair site comparisons

(Table 2), we undertook the reconstruction of soil C storage progression in the 0 to 30 cm soil
layer since the land use conversion from native grassland to annual cropland and subsequently
into shelterbelt (Fig. 3). After normalizing all cropland-chronosequence data (i.e., zero to one;
dimensionless), turnover rates (k) and first-order kinetic models of soil C storage were estimated
(Fig. 3). This long-term k model of soil C depletion in cropped soil had a reassuring coefficient
of determination ($R^2$) of 90% and a very low RMSEn of only 3.34%, which collectively indicates
the high precision of the k model. Over 250 years of cropland chronosequence, the C turnover
rate was quantified as 0.010 ± 0.004 years$^{-1}$, which is equivalent to a MRT of 100 years and a
half-life of soil C of 69 years. This first-order trajectory of soil C depletion in croplands
indicated that 28.9% of the initial soil C under native grassland was gradually lost − i.e., very
likely to the atmosphere − over 250 years of annual cropping (i.e., from 1 to 0.711, Fig. 3). In
further details, during the first year of cropping, we estimated that 0.310% of the pre-existing C
was lost from the soil. Conversely, after 250 years, during the last year of the cropland-
chronosequence trajectory, soil C losses were only 0.026% of the initial soil C – this is one order
of magnitude lower than calculated for the first year of cropping. This deceleration in SOM
mineralization while approaching a new equilibrium and at near steady state was captured
reasonably well by first-order kinetics. Based on the soil C initially present under native
grassland soils (3-chronosequence mean= 125.3 Mg C ha$^{-1}$, Fig. 2), these values of 0.31% and
0.026% were equivalent to C outputs of 392 and 33 Kg C ha$^{-1}$ yr$^{-1}$, respectively.

We projected two potential trajectories (A and B) of how afforested soils can restore soil

C storage in cropland soils over five decades (Fig. 3). After replenishing 81% of the C lost
during long-term annual cropping, the shelterbelts had 94.5% of the initial C of the native
grassland (i.e., $0.81 \times 0.289 + 0.711 = 0.945$; Table 2). Trajectory A was estimated on the basis
that afforested soils fully reached a new steady state and equilibrium of soil C storage with first-
order modelling. This trajectory showed a steep increase in soil C storage over the first decade of
tree planting. In fact, the C accretion rate for trajectory A was 0.119 years$^{-1}$, which suggests a
potential for high soil C accretion under fast C cycling with afforestation. Because the soils
beneath the shelterbelt can still be actively accruing C even 55 years after tree planting, we also
developed trajectory B, which targets a scenario where soil C storage reached 95% of a
theoretical equilibrium (Fig. 3). For this trajectory, the resultant C accretion rate was 0.0334
years$^{-1}$, which corresponds to a modelled MRT of 30 years. When focusing on these progressive
gains of new soil C under trajectory B, the estimated time for half of this soil C portion to enter
the soil was 21 years. Over the two last decades of this progression, the soil C accretion starts to
gradually become asymptotic. During the first year of trajectory B, the net accrual of soil C was
equivalent to 0.88% of the soil C initially present in the native grasslands (Fig. 3). Conversely,
during the last year of trajectory B, soil C accretion corresponded to only 0.17%. Based on the
soil C initially present under native grassland soils (3-chronosequence mean= 125.3 Mg C ha$^{-1}$,
Fig. 2), these 0.88% and 0.017% values were equivalent to a sizable 1.10 and 0.21 Mg C ha$^{-1}$ yr$^{-1}$
$^{1}$, respectively. It is noted that in each of the two afforestation trajectories of soil C accretion (A
and B), the annual contributions of afforestation to net accrual of soil C began from the soil C
storage estimated by the k-modelled cropland trajectory 50 years prior to soil sample collection
(i.e., 0.728, Fig. 3). This is because we had assumed a lag phase of 5 years as noted above, and
tree planting was 55 years prior to soil sample collection.

Based on the assembled k models of cropland-C turnover and simultaneous tree-C

accretion (Fig. 3), of the whole soil C measured beneath the trees at the time of soil sample
collection (Table 2), 25% was estimated to be derived directly from tree-C contributions [i.e.,
$(0.945 - 0.711) / 0.945$]. This can indicate that although tree-C contributions were substantial,
the majority of the C stored in these steppe soils still originated from the initial native grassland
before land use conversion to annual croplands.
**3.2 Sources and turnover of soil C in afforested croplands: United States cases**

Larger accumulation of soil C was consistently found beneath trees relative to the

adjacent annual croplands in all 3 study sites within the Northern Great Plains of the United
States. At the shelterbelts in Norfolk (Fig. 4E) and Mead (data not shown) as well as the forest
plantation in Huron (Fig. 4F), the stable isotope approach followed by mass balance effectively
allocated and quantified the tree-derived soil C, in particular within the 0 to 15 cm soil depth
increment (Table 3).

At the Norfolk site, soils collected from the 0 to 15 cm depth increment beneath the trees

(i.e., within 10 m distance from the center of the shelterbelt) resulted in more than double of the
C mass density found in the annually-cropped topsoils that were located farthest from the trees
(28 vs. 13 Mg C ha$^{-1}$; Table 3, Fig. 4A). Concurrently, when comparing the same surface layer
and spatial sampling locations, soil $\delta^{13}C$ sharply shifted from a considerably depleted -25.7 ± 0.1
‰ beneath the trees to -17.5 ± 0.1 ‰ in the cropped soils north from the shelterbelt (Fig. 4C). As
a result, a significant 79% of the soil C storage measured beneath the trees at the time of sample
collection was attributed to tree-C contributions (Table 3). This translated into a substantial
magnitude of 22 Mg C ha$^{-1}$ being derived specifically from tree biomass (Table 3). Moreover, as
stated above (Method section 2.3), the rest of the soil C was attributed to remaining prairie-C.
The existing soil C beneath trees in the Norfolk shelterbelt allotted to remaining prairie-C was
only 45.5 ± 0.3 % of the whole soil C typically found in the adjacent annual crop field (Table 3).
This indicated that 54.5% of the soil C (equivalent to 13.0 - 5.9 = 7.1 ± 0.4 Mg C ha$^{-1}$, Table 3)
that existed under the long-term annual cropland (i.e., assumed to be at steady state) prior to tree
planting has been lost from the topsoil over the 70 years of afforestation at Norfolk. This net
decline in remaining prairie-C is attributable to $CO_2$ respiratory losses from enhanced biological
activity beneath the trees that gradually accessed, mobilized, cycled and partly mineralized this
legacy prairie-C pool. These results indicated that the turnover of remaining prairie-C in
afforested soils can be even faster than in open cropland fields.

Because the crop field south from the shelterbelt in Norfolk was dedicated to perennial

cropping of alfalfa forage – a C3 species, we undertook a mass balance to distinguish and
allocate the C sources as alfalfa-C vs. remaining prairie-C, with a similar approach as in the
afforested areas (i.e., Eq. [4] and [5]). Within the 0 to 15 cm depth increment, soil C storage
under alfalfa (i.e., 13.1 ± 1.0 Mg C ha$^{-1}$) was the same as in the annual cropland on the north side
of the shelterbelt (Fig. 4A; Table 3). However, the soil $\delta^{13}C$ shifted to -21.6 ± 0.4 ‰, which
resulted in a 41.2 ± 0.4% replacement of the whole soil C storage being derived specifically from
recent contributions of alfalfa-C in this perennial forage field.

Similar to Norfolk, the shelterbelt at Mead also showed a major contribution of

afforestation to whole soil C storage between tree rows in the 0 to 15 cm depth increment (i.e.,
17 Mg C ha$^{-1}$, Table 3), which corresponded to 37% of the whole soil C. It is noted that although
the magnitude of original prairie-C lost after 35-yr of afforestation at Mead (i.e., 36.2 - 29.3 =
6.9 ± 0.7 Mg C ha$^{-1}$, Table 3) was comparable to Norfolk, the proportion of this original prairie-
C lost from afforested soils in Mead was much smaller with only 19% (i.e., 100 – 80.9; Table 3).
This can indicate that the net changes in whole soil C storage under afforestation (i.e.,
simultaneously encompassing the noted prairie-C losses and the asymmetrically-larger tree-C
gains) did not follow a fixed proportionality to the initial prairie-C.

At the 19-year-old forest plantation at Huron, trees also increased soil C storage while

decreasing the $\delta^{13}$C signature, in particular in the 0 to 15 cm depth increment (Fig. 4B, Fig. 4D).
When contrasting the afforested soils collected at least 17 m away from the plantation boundary
vs. topsoils taken within the adjacent cropland from the sampling locations that were farthest
removed from the trees, $\delta^{13}$C changed from -20.7 ± 0.2 ‰ to -17.3± 0.1 ‰ ‰, respectively.
Although trees were much younger in Huron than in Mead and Norfolk, a considerable
magnitude of tree-C was found (i.e., 12 Mg C ha$^{-1}$, Table 3). This indicated that direct tree
contributions to soil C storage can take place rather quickly, within a few decades following
afforestation. However, the changes in the remaining prairie-C pool beneath the trees at Huron
apparently differed from what was found in both Mead and Norfolk. While soils beneath trees at
both Mead and Norfolk showed declines in remaining prairie-C relative to whole soil C stored in
open cropland fields at the time of sample collection, the afforested soil at Huron showed no
change in the magnitude of remaining prairie-C in the 0 to 15 cm depth increment. In fact, it is
striking how similar the whole C in the cropped soil ($25.6 \pm 0.7$ Mg C ha$^{-1}$) was to the C
allocated to the remaining prairie-C pool beneath the trees ($25.3 \pm 0.7$ Mg C ha$^{-1}$, Table 3). What
is more, the soils sampled from the adjacent native grassland within the Huron site also returned
a very consistent magnitude of soil C storage, with $25.4 \pm 1.9$ Mg C ha$^{-1}$ (n= 3, data not shown).
Provided the uncertainty of field sampling, this evidence strongly indicated that all or nearly all
the original prairie-C was retained and still present in the Huron soils under both annual cropping
and afforestation (Table 3). Likewise, when examining the 0 to 30 cm soil depth increment at
Huron (i.e., aggregating the two sampled soil layers shown in Fig. 4B), we further corroborated
this similarity in soil C storage between native grassland and annual cropland ($45.4 \pm 1.0$ vs. 44.2
$\pm 1.1$ Mg C ha$^{-1}$, respectively, data not shown)
Significant regressions revealed the consistent dependency of C source allocations on
time since tree planting (Fig. 5). Over time following afforestation, tree-C source increased
linearly from an assumed null contribution at planting to become 79% of the whole soil C after
70 years of tree planting in Norfolk ($R^2= 0.95$, Fig. 5A). We also evaluated changes over time for
the remaining prairie-C in afforested soils relative to the corresponding adjacent croplands within
each study site. Linearity of these prairie-C proportions as a function of time was also observed
when encompassing the 3 study sites ($R^2= 0.999$, Fig. 5B). As described above, the more
recently-afforested soils at the Huron site kept the entire prairie-C, while the oldest afforested
soils at the Norfolk site retained less than half of the whole soil C present in the adjacent annual
croplands (45.5%).
We focused on estimating the turnover rates of soil C mass density derived directly from
tree-C sources while encompassing the range of conditions in the 3 studies. As most of the
beneficial effects of tree planting across the sites in the United States were detected in the 0 to 15
cm soil depth increment, further examination of accretion rates of soil C storage focused on this
specific topsoil layer. Upon assembling the magnitudes of tree-C contributions over time since
afforestation, unified first-order kinetics modelling converged and emerged robustly ($R^2$= 0.997,
Fig. 6). The k rate constant of 0.0552 years$^{-1}$ corresponds to a half-life of 12.6 years, which
indicates that more than half of the accrual tree-C occurred within less than two decades (when
accounting for a lag phase of 5 years following tree planting) (Fig. 6). This further substantiated
the rapid contributions of afforestation to increase soil C storage quickly until reaching a new
dynamic equilibrium. This generalized relationship enabled projecting tree-C accruals in
afforested soils within the assessed time range of 70 years. We further implemented this robust
k-progression to simultaneously depict the gains in tree-C while also representing the declines in
prairie-C in afforested soils for each study site separately (Fig. 7). This approach accounts for the
C that is being lost from net mineralization of pre-existing C in the remaining prairie SOM (Fig.
7). It was noticeable that the afforested soils at Mead showed faster turnover rate of the
remaining prairie-C than the other two sites by approximately two-fold. The k rate constant of
net mineralization of prairie-C beneath trees at Mead was 0.145 years$^{-1}$ (Fig. 7B), which
corresponded to an MRT of about 7 years. This implies that the average time for prairie-C to be
lost from Mead afforested soils was well within one decade, whereas prairie-C in afforested soils
in Huron and Norfolk showed longer residence times by about double.

**4 Discussion**
**4.1 Carbon contributions from trees to SOM sequestration**
Planting trees in croplands creates substantial sinks of atmospheric C in the soil profile
(Sauer et al., 2007; Khaleel et al., 2020; Zhang et al., 2020). Current knowledge of this important
benefit of afforestation has been deepened and reinforced earlier literature (Post and Kwon,
2000; Hernandez-Ramirez et al., 2011; Chendev et al., 2015b). It is noticeable that having long-
term annual croplands as the land use system prior to establishing trees particularly enlarges the
soil C sink and replenishment caused by afforestation (Guo and Gifford, 2002; Laganiere et al.,
2010; Sauer et al., 2012). Overall results indicate that across tree species and local edaphic-
climatic conditions at the studied sites, the massive tree-C contributions through decaying roots
and litter (Li et al., 2012; Amadi et al., 2016) can saturate the soil with C substrates in surplus to
the capacity of microbial decomposition (Li et al., 2018; Deng et al., 2014), which collectively
incline the C balance towards net C accrual (Hernandez-Ramirez et al., 2011). Our quantification
of these tree biomass-C contributions to newly-accrued soil C further expand this growing body
of knowledge. The proportion of new soil C originated from tree biomass were shown to increase
significantly with time (Fig. 5A).
By contrast to afforestation, long-term annual cropping implies recurrent soil mixing,
disruption of any preexistent vertical stratification, microclimate fluctuation, and exposure of
SOM to decomposition, which collectively shift the C balance and predispose towards depletion
of soil C (Post and Kwon, 2000; Hernandez-Ramirez et al., 2009; Curtin et al., 2014). In addition
to this disturbance and exposure of SOM to decomposition, low C inputs and high C removals
via harvest are also characteristic of conventional annual cropping systems. Moreover, results
indicated that C from crop residues was largely lost back to the atmosphere every year with no
significant net contribution to the soil organic C pools in the long term (Fig. 2, Fig. 3). A
reduction of C inputs in croplands once the native vegetation (roots and aboveground biomass)
have been removed (Hu et al. 2013) is also typically followed by alterations in soil physical
properties such as decreases in porosity and gas exchange which can become detrimental to plant
primary productivity and soil biology (Kiani et al., 2017).
Our study explicitly examined and quantified for first time in literature the losses of
remaining prairie-C directly beneath trees across afforested soils (Fig. 3, Fig. 5B, Fig. 7). This
analysis showed that under afforestation, soil C remaining from original native grasslands
continues to be lost from the profile, likely via microbial mineralization (Fig. 3, Fig. 7). It is
noted that the accretion of recently-added tree-C is much faster than these observed losses of
remaining prairie-C beneath trees as the recently-added plant-C is considered relatively more
labile than prairie-C. The noted decline in remaining prairie-C beneath young afforestation
agrees well with a decomposition of SOM in the early stage of tree growth as previously
deliberated by Paul et al. (2002), Garten (2002), and Xiong et al. (2020). At the Norfolk site,
tree-C contributions effectively replenished and greatly surpassed the gradual losses of
remaining prairie-C in the soil (Fig. 7C). In the case of the Huron site, afforestation conserved
the initial prairie-C while also contributing directly to additional tree-C accrued in an overall
increasing SOM pool (Fig. 7A)
It is noted that although the 3 US sites (Norfolk, Huron and Mead) shared a common
trajectory of tree-C accretion with time (Fig. 5, Fig. 6), their k turnover rates of remaining
prairie-C differed (Fig. 7). These apparent divergences are potentially attributable to differences
in temperature and moisture regimes across the region. In further details, in the case of accretion
of tree-C in afforested soils, this response to land use change seems governed mostly by the
change into tree vegetation and the duration of afforestation; therefore, it became feasible for us
to establish a unified, robust k model across a range of afforestation ages (Fig. 6). Conversely,
loss rates of remaining prairie-C in afforested soils appeared to be mostly contextual and even
site specific, likely as a function of local climatic conditions (Chendev et al. 2014, 2015b).
Relative to both Norfolk and Huron (Fig. 7), warmer-wetter conditions in Mead could have led
to the faster C turnover rate and mineralization of the remaining prairie-C in these afforested
soils (Fig. 7B). Overall, these results exemplify how analyzing the compartments of soil C
turnover – evaluating separately tree-C contributions and remaining prairie-C, instead of
studying only the whole soil C – can provide further insights into SOM dynamics following land
use conversions. Future research could address the potential existence of underlying thresholds
of heat and moisture availabilities that are conducive to retain and converse pre-existing prairie-
C in afforested soils while simultaneously enabling soil C accretion directly from new tree-C
contributions. Likewise, SOM fractionation approaches offer excellent avenues for further
unravelling the stabilization mechanisms of C in the soil.

Based on the kinetics-modelled reconstruction of soil C storage over time in the Russian

land use chronosequences (i.e., encompassing a range of different ages since conversion to
annual cropping), over the 55 years that elapsed since tree planting until soil sample collection,
the remaining soil C from the original native grassland was shown to be lost continually (Fig. 3).
Our turnover estimations using kinetics modelling suggested that only 1.7% of the initial
grassland-C was lost over these 55 years following shelterbelt afforestation (Fig. 3). Based on
these results from the Russian chronosequences, the relatively small grassland-C loss is in part
because soil C had been depleted over nearly two centuries of annual cropping prior to tree
planting. Nevertheless, mycorrhizae activity in afforested soils can preferentially access and
utilize remaining grassland-C beneath trees (Mellor et al., 2013). Hence, this biological effect
could contribute to gradual decreases in remaining grassland-C in afforested soils.
Chendev et al. (2014, 2015b) further addressed differences in soil C accrual across
afforested sites in Russia and the United States, also attributing them primarily to differences in
moisture regimes. Within each geographic region as well as in the collective of both countries,
they explained that cooler-moister conditions led to increases in overall soil C accrual beneath
trees. This postulate is clearly in line with earlier results by Garten (2002). Potential increases in
plant primary productivity with increasing moisture as well as reductions in microbial
mineralization of the overall SOM with colder conditions can shift and drive the C balance in the
soil towards net C accrual.
Of the 3 paired sites in Russia (Table 2), the Yamskaya site is probably the most
representative and closely related to the 3 long-term chronosequences evaluated in this study.
This is because Yamskaya and the 3 chronosequences are all geographically located within the
Belgorod oblast, and hence, they share a more similar regional climate. It was striking that the
afforested soil at Yamskaya site had a soil C accretion even greater than the native grassland
reference, which strongly indicated the high capacity of shelterbelts to sequester C even beyond
the capacity of the corresponding native ecosystem (Table 2). After noticing this finding, it can
also be anticipated that although the drier Kamennaya site showed at the present the slowest soil
C accretion following afforestation of SOM-depleted croplands (i.e., 5% of C restoration; Table
2), it is possible that in the long term, this drier environment can gradually sequestered even
more soil C than the moist sites located in Belgorod oblast (e.g., Yamskaya). This is suggested as
the Kamennaya site exhibited the highest soil C storage when comparing across all the native
grasslands compiled in our study (i.e., 152.5 Mg C ha$^{-1}$; Table 2 and Fig. 2).
**4.2 Turnover rates of soil carbon as a function of land use changes**
This study clearly confirms that the long-term dynamics of soil C is consistently
nonlinear, either during decline or accumulation of soil C as a function of land use choices. As
deliberated earlier by Post and Kwon (2000) and Garten (2002), erroneously assuming linearity
in depicting these trajectories of soil C would lead to underestimating the rates of soil C changes
during the first decades following a land use conversion as well as overestimating the turnover
rates of soil C after multiple decades once the ecosystem has actually reached stability and a
balance between their C inputs and outputs. This latter notion essentially applies when long-term
cropland or afforested fields have become mature (Hernandez-Ramirez et al., 2011)
Earlier chronosequence and stable isotope analyses by Arrouays et al. (1995) in
Southwest France further support that land use effects on soil C changes take place rather
quickly. They reported that a new equilibrium in soil C storage was reached within only few
decades of a land use change from forest to annual croplands, and with about half of the C loss
occurring rapidly within few years (< 10) of beginning cultivation (Arrouays et al., 1995).
Similarly, as in our study, only a few decades seems to be required to reach equilibrium when
switching from cropland to trees (Richter et al., 1999; Paul et al., 2002; Guo and Gifford, 2002).
Likewise, comprehensive results by Dhillon and Van Rees (2017) depicting soil C accretion
caused by afforestation in the Canadian Prairies can be interpreted as net C losses taking place
over the first several years after tree planting, and subsequently, an ensuing fast accrual of soil C
until tree ages of about 35 years when new equilibria or C sequestration ceilings under
afforestation can be reached. In our study, MRT of soil C beneath trees were in general
determined to be about two decades (Fig. 6). Furthermore, the evaluation of our two scenarios of
asymptotic equilibria of C accrual in afforested soils (i.e., trajectories A and B in the normalized
Russian chronosequences; Fig. 3) can provide the boundaries of faster vs. slower accretion rates
of soil C with the corresponding MRTs of one vs. three decades.
In the case of long-term annual croplands in Russia, the soil C MRT of 100 years found
in our study (i.e., associated with a k rate constant of 0.010 years$^{-1}$, Fig. 3) is comparable to
findings by Huggins et al. (1998) who registered MRTs of 91 and 143 years in annual cropping
systems in Minnesota, but overall longer than a report by Collins et al. (1999) who found a wide
range of MRTs between 18 to 96 years for sites with 8 to 33 year-old continuous maize cropping
across the Central United States, respectively.
It is noted that the exponential first-order trajectory of soil C turnover in the Russian
chronosequences (Fig. 3) was captured better with Eq. [1] than the simplistic $C_{(t)} = C_o \times e^{-kt}$
previously used by Hernandez-Ramirez et al. (2011). While Eq. [1] provided an $R^2$ of 90% (Fig.
3), $C_{(t)} = C_o \times e^{-kt}$ returned an $R^2$ of 68% (data not shown). With two fitting parameters (i.e., $C_o$,
$C_e$), first-order kinetic modelling with Eq. [1] represented reasonably well the assumptions of
steady state and new equilibrium at the end of the evaluated time series (Fig. 3, Fig. 6).
Further kinetics modelling efforts of soil C increases in afforested systems can take the
form of two functional C pools where inputs and outputs to labile and recalcitrant SOM can be
predicted (Arrouays et al., 1995; Garten, 2002; Hernandez-Ramirez et al., 2009; Xiong et al.,
2020). Preferentially accruing C into recalcitrant vs. labile SOM pools in afforested soils can be
interpreted as tree-C contributions towards long- vs. short-term stability of soil C storage,
respectively, with crucial ramifications for mitigation of future climate change (Laganiere et al.,
2010; Hernandez-Ramirez et al., 2011; Deng et al., 2014). Future investigations can also focus
on the protection and stabilization mechanisms of SOM as created by soil aggregate formation
beneath trees (Kiani et al., 2017; Quesada et al., 2020). Once soils subjected to long-term annual
cropping are converted to permanent vegetation, fungal hyphae can become an important means
that mediates C accretion by enhancing soil aggregation (Jastrow et al., 1996; Kiani et al., 2017).
Jastrow et al. (1996) indicated that fungal hyphae could improve macroaggregation and hence
indirectly enhance C accrual. Using phospholipid fatty acid biomarkers, Kiani et al. (2017)
identified a linkage between presence of fungal biomass and increases in hierarchical fractal
aggregation specifically in forest soils, while this association was absent in the adjacent
annually-cropped soils in their study. Furthermore, Quesada et al. (2020) recently discussed the
mechanisms for soil C accretion in tropical forests. In line with earlier findings by Wang et al.
(2016), Quesada et al. (2020) stated that SOM physical protection provided by the formation of
soil aggregates slows decomposition of SOM within aggregates, and hence, it becomes a second
layer of stabilization after realizing the primary SOM stabilizing effects caused by mineral
surfaces of fine soil particles such as silt and clay. Further studies can focus on the effects of
inherent mineralogy and texture as well as clay lessivage processes on C dynamics and storage in
afforested soils (Chendev et al., 2020; Quesada et al., 2020).

**5 Conclusions**
Nonlinear turnover rates of soil C revealed an MRT of a century in long-term croplands
as soil C slowly undergoes depletion and losses to the atmosphere. Likewise, when croplands
were afforested, nonlinear accretion rates of soil C indicated a MRT of approximately two
decades following afforestation. Soil C showed to be rapidly accrued as trees remove $CO_2$ from
the atmosphere and contribute C substrates for SOM accumulation and stabilization. While our
study confirmed these substantial C accruals in the soils under the trees, the overall gain at the
actual landscape scale will depend in part on the proportion of farmland dedicated to
afforestation, with afforested areas typically accounting for up to 5% of the farmlands (Amadi et
al., 2016).

Our focus on soil organic C behavior in soils under shelterbelts is only part of a broader

range of studies evaluating the overarching impacts of agroforestry on soil quality and crop yield
across the landscape. Beyond C sequestration, the benefits of shelterbelts can also be manifested
in improving the local climate as well as increasing crop yields. Results collectively
substantiated that in addition to multiple benefits by trees such as providing air quality,
microclimate regulation and erosion control (Sauer et al., 2007; Hernandez-Ramirez et al., 2012),
C sequestration in afforested lands is a suitable means to proactively address and effectively
mitigate ongoing climate change within a person's lifetime.




**Data availability**

Data are available on request.

**Author contribution**

GHR: Conceptualization; Methodology; Formal analysis; Funding acquisition; Resources; Visualization; Writing the original draft; Review and editing new versions; Corresponding author role.

TJS and YGC: Conceptualization; Data curation; Formal analysis; Funding acquisition; Investigation; Methodology; Supervision; Project administration; Resources; Visualization; Review and editing new versions.

ANG: Investigation; Methodology; Funding acquisition; Review and editing new versions.

**Competing interests.**

The authors declare that they have no conflict of interest.

**Acknowledgement**

The authors are grateful for the valuable assistance provided by Kent Heikens, Kevin Jensen, David Denhaan, and Amy Morrow. The first author acknowledges early encouragements and beneficial exchange of ideas on the study subject with Dr. Cindy A. Cambardella − now sadly deceased. We sincerely appreciate the funding support as follows: from Natural Sciences and Engineering Research Council of Canada (NSERC) Discovery program (2018-05717) as well as Alexander von Humboldt Foundation (Germany) (CAN - 1206917 - HFST-E) to GHR; from U.S. Civilian Research and Development Foundation - Cooperative Grants Program (Project

RUG1-7024-BL-11) to TJS and YGC; and from Russian Science Foundation (Project 19-17-
00056) for field sample collection and laboratory analyses within Russian sites to YGC and
ANG.

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

Table 1. Topsoil pH and textures at the 9 study sites. Numerals correspond to sites as shown from west to east in Fig. 1.

| Site | Soil pH | Soil texture |
| --- | --- | --- |
| 1. Huron† | 7.0 | sandy loam |
| 2. Norfolk† | 6.8 | loamy sand |
| 3. Mead† | 6.1 | silty clay loam |
| 4. Streletskaya Steppe‡ | 7.0 | loam |
| 5. Ivnyanskiy‡ | 7.5 | silt loam |
| 6. Prokhorovskiy‡ | 7.2 | loam |
| 7. Gubkinskiy‡ | 7.4 | clay loam |
| 8. Yamskaya Steppe‡ | 7.2 | loam |
| 9. Kamennaya Steppe‡ | 7.6 | clay loam |

† correspond to the cropland location within this site.
‡ correspond to the native grassland location within this site.

Table 2. Soil organic carbon storages and differences within the 0 to 30 cm depth increment under three land uses (i.e., native grasslands, annual croplands and afforestation) in Russia. The nine values of soil C storage across the nine site-land uses were previously presented and discussed in Chendev et al. (2015b) and are repeated here for informing first-order kinetic modelling and estimations of C accretion rates when converting from annual croplands to afforestation as shown in Fig. 3. At time of soil sample collection, the ages of the annual croplands in Streletskaya and Yamskaya were at least 140 years, and at least 145 years of age in Kamennaya. In all 3 sites, the shelterbelts had been planted 55 years prior to soil sample collection.

| Land use or descriptor | Streletskaya Steppe site, Kursk | Yamskaya Steppe site, Belgorod | Kamennaya Steppe site, Voronezh | 3-sites mean | Standard error |
|---|---|---|---|---|---|
| | Soil C mass density (Mg C ha$^{-1}$) | | | | |
| Native grassland (G) | 126.2 | 138.0 | 152.5 | 138.9 | 7.61 |
| Annual cropland (C) | 109.3 | 127.2 | 123.6 | 120.0 | 5.47 |
| Shelterbelt (Trees) | 126.4 | 142.1 | 125.0 | 131.2 | 5.48 |
| Net decrease G-to-C | -16.9 | -10.8 | -28.9 | -18.9 | 5.32 |
| Net increase C-to-Trees | 17.1 | 14.9 | 1.40 | 11.1 | 4.91 |
| G-to-C / C-to-Trees† | 1.01 | 1.38 | 0.05 | 0.81 | 0.40 |

†Ratio representing the replenishing of depleted soil C by tree planting. These ratios were calculated as the absolute values of net increase from cropland to shelterbelt (trees) divided by net decrease from grassland to cropland.

Table 3. Soil organic carbon storage within the 0 to 15 cm depth increment contrasting annual croplands and afforestation in United States. Sources of soil C storage beneath the trees were allocated as tree-C versus remaining prairie-C using Eq. [4] and [5] and associated assumptions. Error bars are standard errors of the means.

| Site | Tree age | Whole soil C mass density | | Tree-C contribution to soil C | | Remaining prairie-C | | Remaining prairie-C / whole C in cropland† |
|------|------|------|------|------|------|------|------|------|
| | (years) | Annual cropland (Mg C ha$^{-1}$) | Beneath trees (Mg C ha$^{-1}$) | Mass density (Mg C ha$^{-1}$) | Proportion (%) | Mass density (Mg C ha$^{-1}$) | Proportion (%) | Proportion (%) |
| Huron, S. Dakota | 19 | 25.6 ± 0.7 | 37.5 ± 1.2 | 12.2 ± 0.8 | 32.6 ± 1.5 | 25.3 ± 0.7 | 67.4 ± 1.5 | 98.8 ± 2.8 |
| Mead, Nebraska | 35 | 36.2 ± 0.4 | 46.7 ± 1.5 | 17.4 ± 1.2 | 37.2 ± 1.8 | 29.3 ± 1.9 | 62.8 ± 1.8 | 80.9 ± 3.6 |
| Norfolk, Nebraska | 70 | 13.0 ± 1.3 | 28.0 ± 1.4 | 22.1 ± 1.1 | 78.9 ± 1.0 | 5.90 ± 0.42 | 21.1 ± 1.0 | 45.5 ± 0.3 |

† This ratio represents the proportion of remaining prairie-C relative to whole soil C in annual cropland. The magnitudes of both 'remaining prairie-C' and 'whole soil C in annual cropland' are shown in other columns of this same table. It is noted that balance (e.g. in Mead, 100 - 81 = 19) represents to the proportion of prairie-C lost in afforested soils since tree planting.

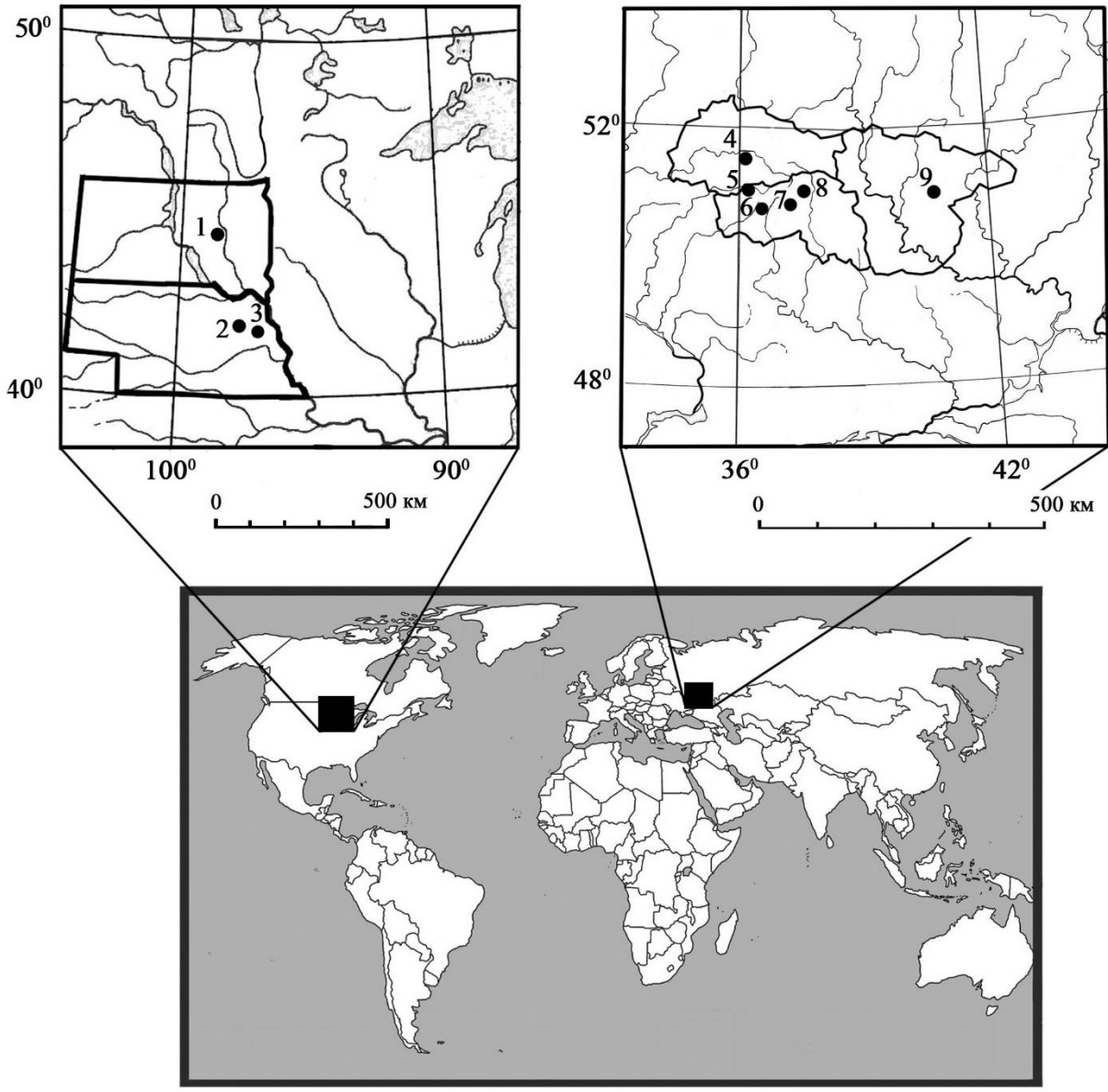

Fig. 1. Geographical location of the nine study sites within the United States (1. Huron, 2. Norfolk, 3. Mead) and Russia (4. Streletskaya Steppe, 5. Ivnyanskiy, 6. Prokhorovskiy, 7. Gubkinskiy, 8. Yamskaya Steppe, 9. Kamennaya Steppe). Within Russia, 5, 6, and 7 are sites with chronosequences of land use conversion, while 4, 8 and 9 are paired sites (native grasslands, annual croplands versus shelterbelts). The 3 sites in the United States are paired sites (afforestation vs. adjacent annual croplands).

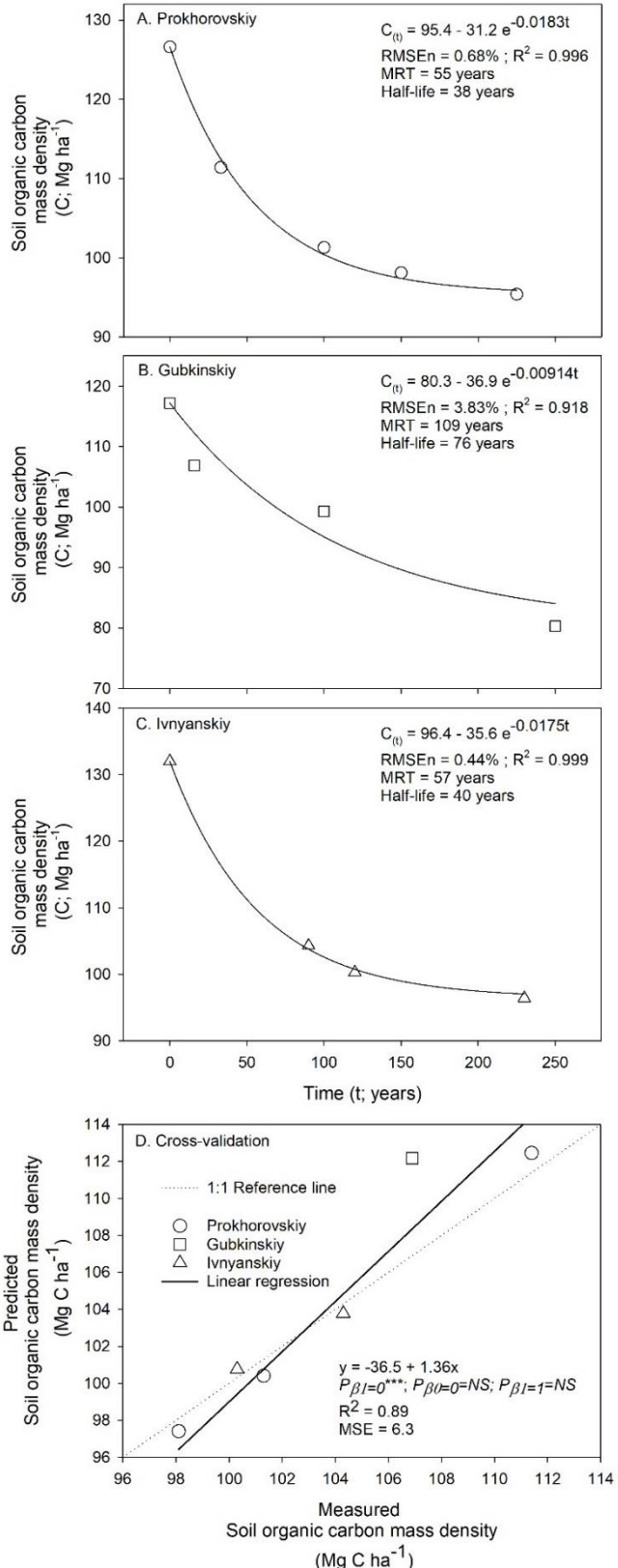

Fig. 2. Land use chronosequences of soil organic carbon storage within the 0 to 30 cm depth increment after converting native grassland to annual croplands in Belgorod oblast, Russia. These showed soil C declines over time. (A) Prokhorovskiy, (B) Gubkinskiy and (C) Ivnyanskiy districts. In Panels A, B and C, first-order kinetic models are described by the solid curvilinear fittings and equations in the form $C_{(t)} = C_e + (C_o – C_e) e^{-kt}$ where $C_e$ is C at new dynamic equilibrium (inputs = outputs), $C_o$ is initial C at time of land use conversion (time zero), k is the first-order kinetic rate constant equivalent to turnover rate. As reciprocal of k, MRT stands for mean residence time, while half-life equates to $\ln(2)/k$. (D) Cross-validation of first-order predicted C versus measured C encompassing the 3 chronosequences within the age range from 10 to 200 years. The subscripts of the *P* values denote the null hypotheses for testing the regression coefficient ($\beta_1$) and intercept ($\beta_0$), where \*\*\* is a p-value of <0.001 and *NS* is non-significant. First-order kinetic modelling was supported by this performance evaluation.

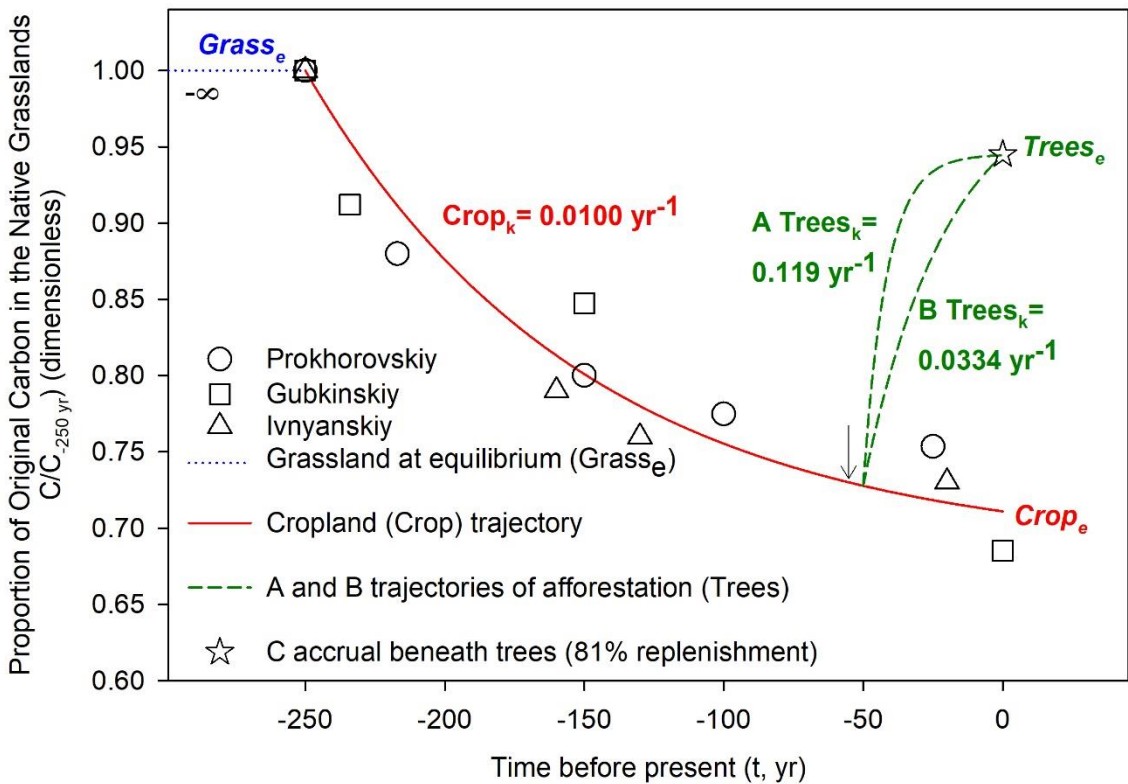

Fig. 3. Reconstruction of soil organic carbon storage within the 0 to 30 cm depth increment following land use conversions from native grassland to annual cropland and subsequently into afforestation with shelterbelts in Russia. This assemblage assumed that native grasslands were at dynamic equilibrium and steady state prior to conversion to annual croplands. Likewise, nonlinear k estimates of turnover rates in croplands and accretion rates under afforestation also required the assumptions of reaching new dynamic equilibria and steady state at zero year (i.e., time of soil sample collection). The cropland trajectory of soil C over time was derived from chronosequence data presented in Fig. 2A, Fig. 2B and Fig. 2C. The soil C accrual beneath trees at time zero (open star) was estimated from measured data presented in Table 2 (i.e., of the soil C that had been depleted by cropping, afforestation replenished 81%, based on 3-sites mean). Note that although trees were planted on year -55 (vertical arrow ↓), tree-C contributions to soil C accrual were accounted for starting from year -50 based on a literature review by Paul et al. (2002) that suggested a lag phase of 5 years. The trajectory 'B Trees' (dashed red line with k= 0.0334 yrs$^{-1}$) assumed that the soil C storage had asymptotically reached 95% of a theoretical equilibrium (i.e., 'Trees$_e$' / 0.95). First-order kinetic modelling was used to derive these three nonlinear trajectories of soil C (Eq. [1]). With the aim of integrating information from the 3 available chronosequences (Fig. 2), all soil C storage data were normalized (i.e., zero to one) and presented here as fractions of C storage at time of conversion from native grassland to annual cropland (shown as -250 years before time of soil sample collections).

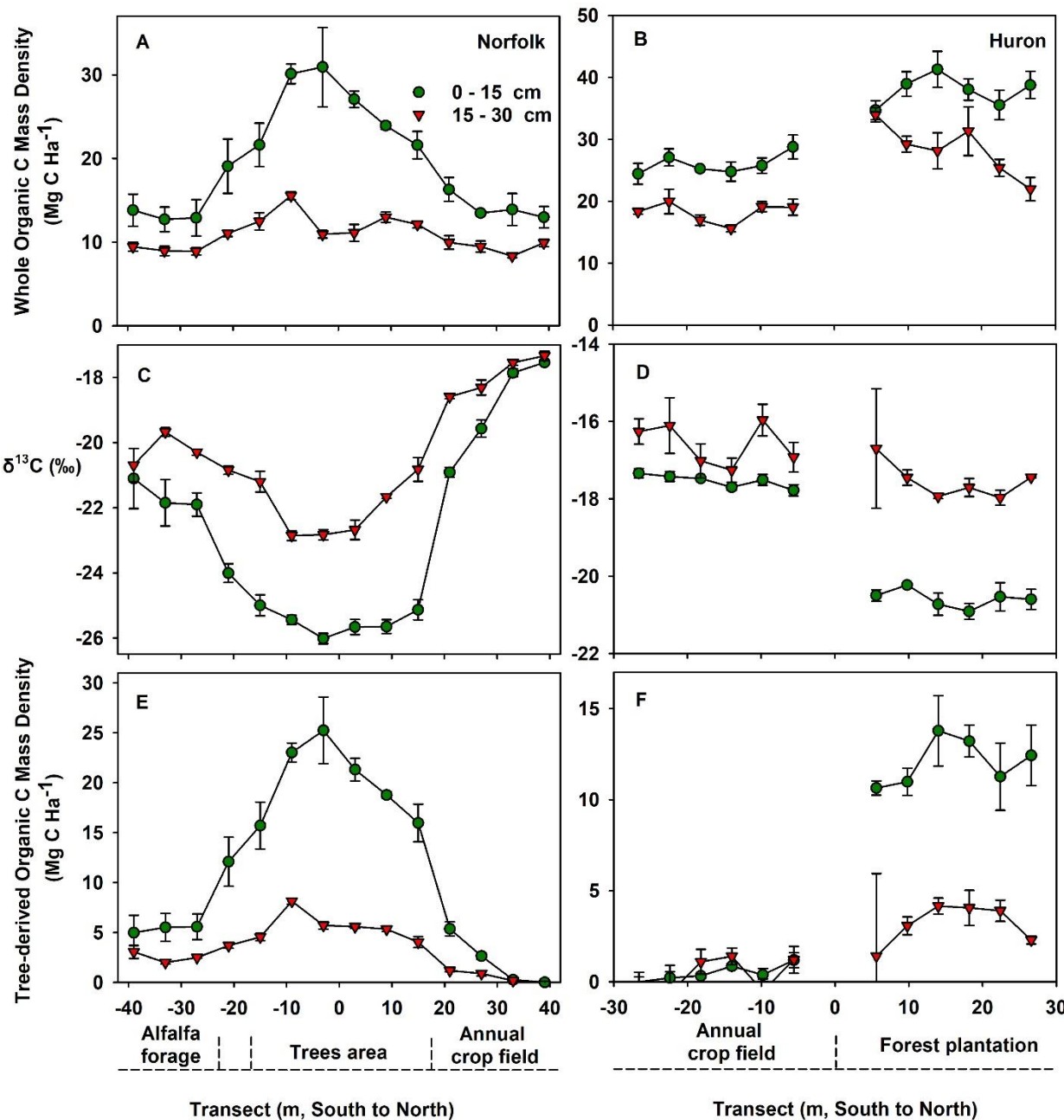

Fig. 4. (A, B) Soil organic carbon mass storage, (C, D) stable isotope ratios of organic C ($\delta^{13}$C), and (E, F) organic C mass derived from C3 plants across transects at Norfolk shelterbelt (left panels) and Huron forest plantation (right panels) for the 0 to 15 and 15 to 30 cm soil depth increments. Adjacent cropped fields were also included. Within the afforested areas in panels E and F, the reported organic C masses are primarily attributed to direct tree contributions. Contributions of tree-C to soil C storage were clearly discernable within the 0 to 15 cm depth increment. Note the difference vertical y-scales across panels. Error bars are standard errors of the means, with sample sizes of 4 for Huron and 3 for Norfolk.

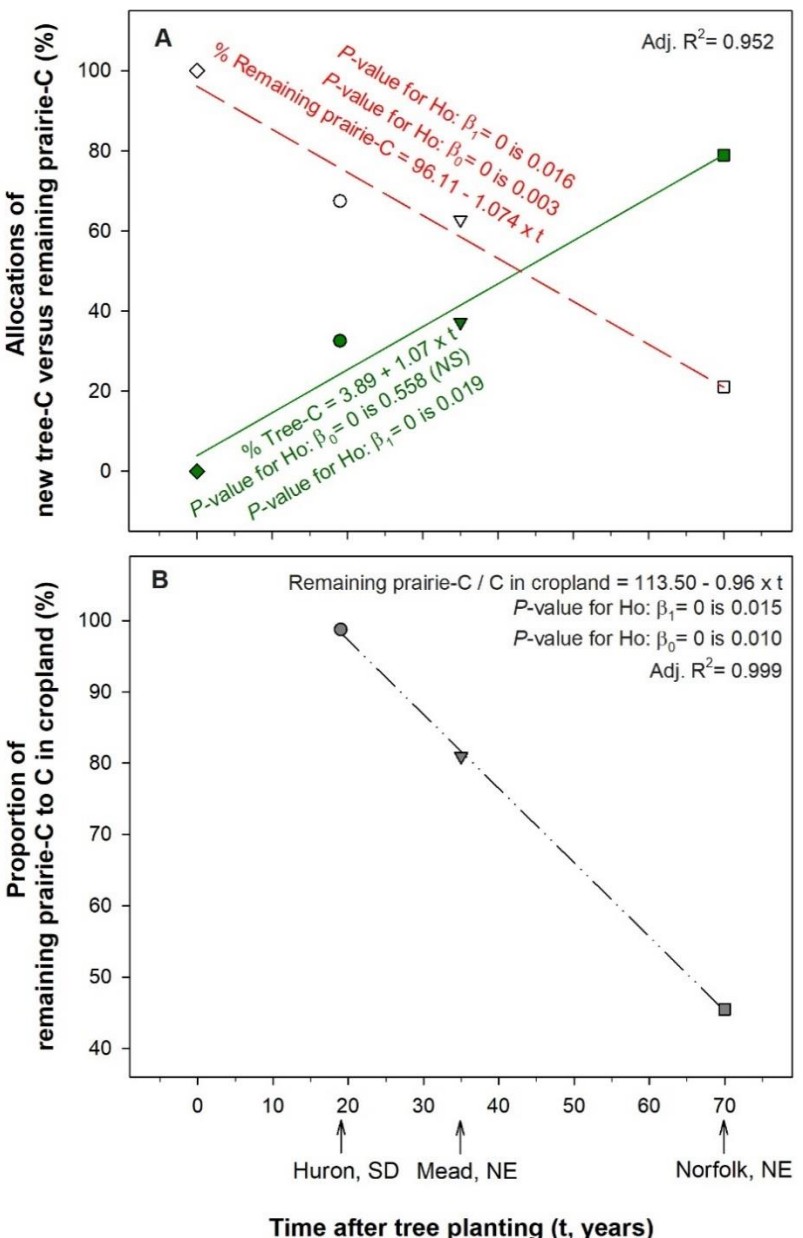

Fig. 5. Proportions of soil organic carbon within 0 to 15 cm depth increment (A) from tree-C versus remaining prairie-C relative to whole soil C stored directly beneath trees, and (B) remaining prairie-C in afforested soils relative to whole soil C in the adjacent croplands. It was inferred that all soil C different from new tree-C was preexisting soil C attributable to remaining prairie-C. (A) Mead data was recalculated from Hernandez-Ramirez et al. (2011) as compiled in Table 3. The tree-C source increased to become 79% of the whole soil C over 70 years after tree planting. (B) The proportion of prairie-derived C beneath trees declined over time to become less than half (45%) of the whole soil C in the adjacent annual croplands, which were assumed to be at steady state.

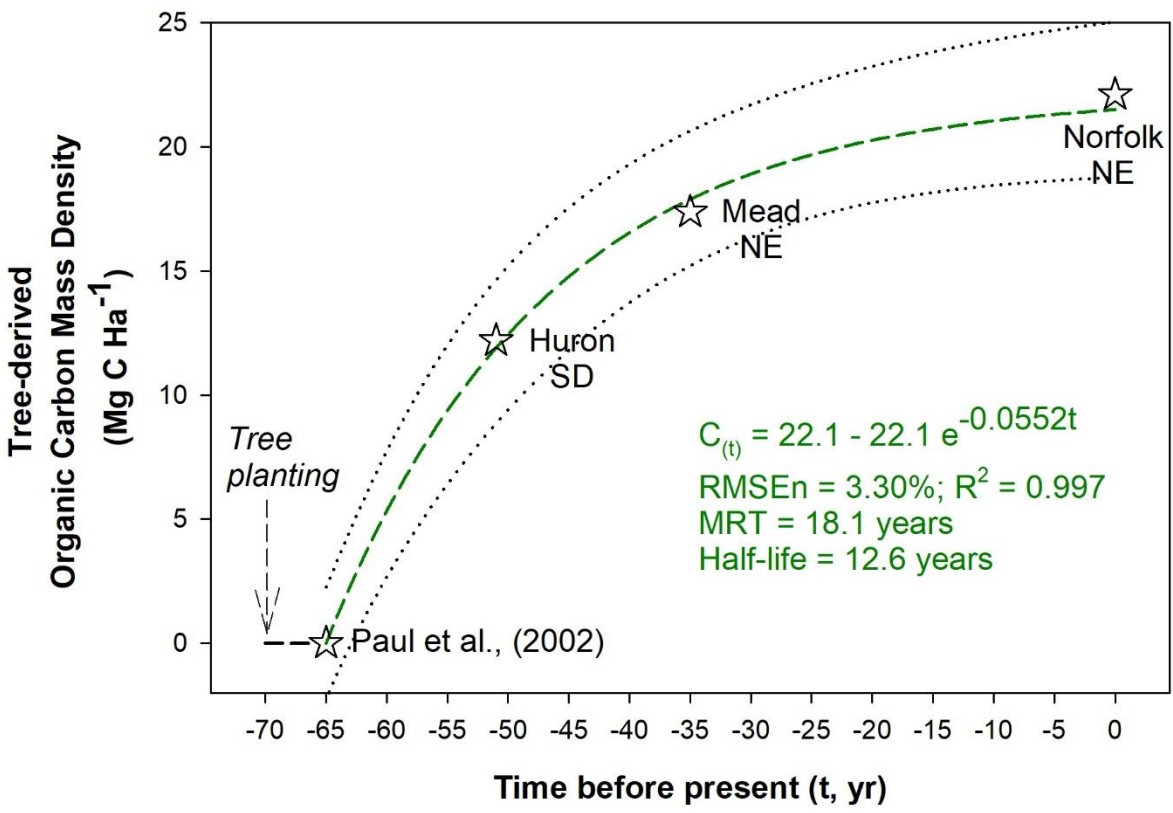

Fig. 6. Reconstruction of soil organic carbon storage within the 0 to 15 cm depth increment following land use conversion from annual cropland to afforestation United States. The arrow indicates the time of tree planting (-70 years). Based on a literature review by Paul et al. (2002), we included a lag phase of 5 years following tree planting. Huron and Norfolk data were derived from results presented in Fig. 4. Mead data was recalculated from Hernandez-Ramirez et al. (2011) as compiled in Table 3. This assemblage supports that afforested soils were approaching steady state at nearly 70 years after tree planting, as required for first-order kinetic modelling. The first-order kinetic model (Eq. [1]) is depicted by the solid curvilinear fitting, where k is the first-order kinetic rate constant equivalent to accretion rate under afforestation, and MRT stands for mean residence time. Normalized root mean square error (RMSEn) and coefficient of determination ($R^2$) for the k model are also provided. The 95% prediction bands of this k model are provided as dotted lines. This nonlinear trajectory describes and highlights the contribution of trees to soil C accrual.

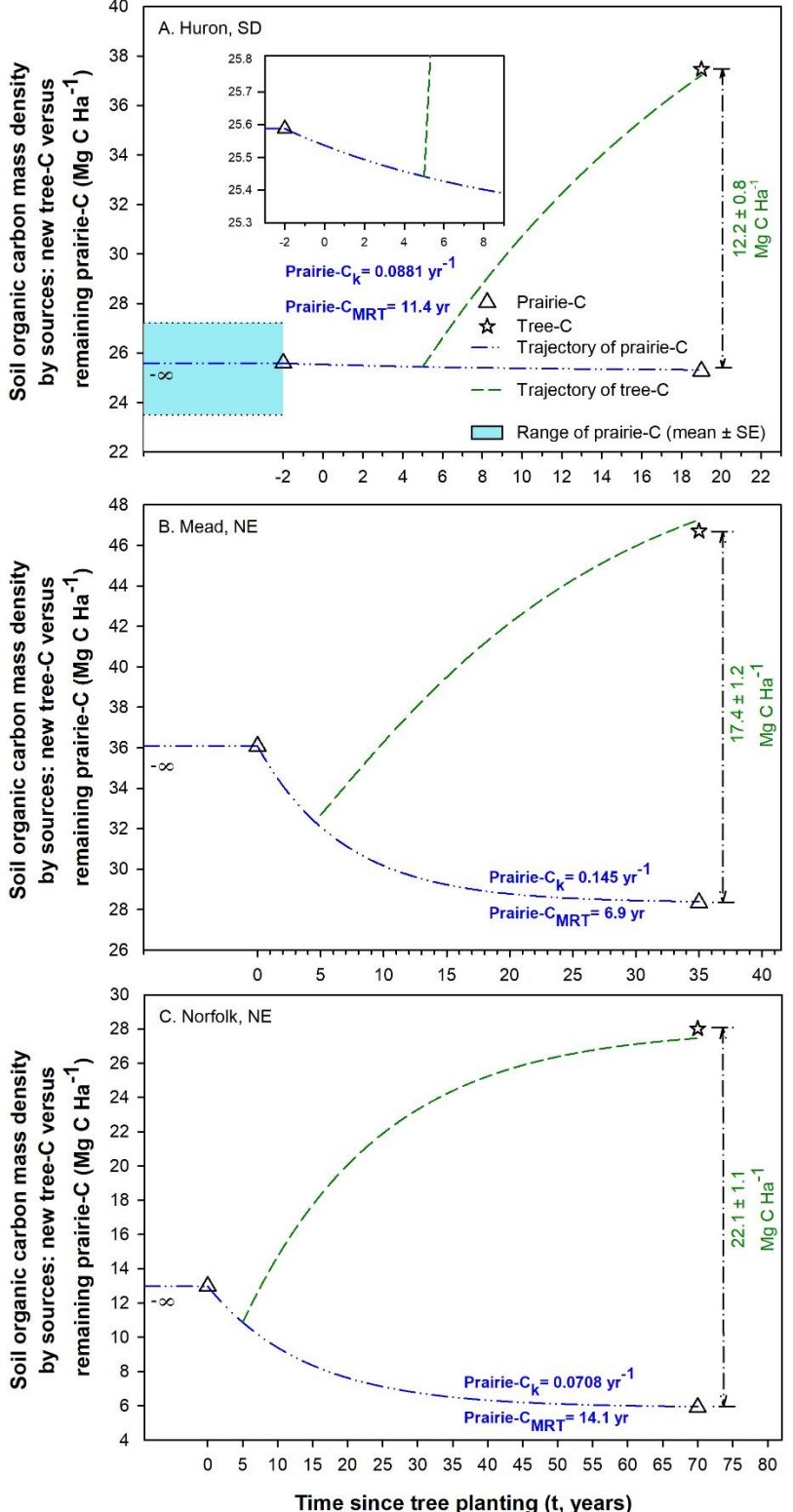

Fig. 7. Net carbon accretion in afforested soils caused by simultaneous C gains from substantial tree-C contributions and smaller C losses from mineralization of remaining prairie-C. It was inferred that all soil C different from new tree-C was preexisting soil C attributable to remaining prairie-C. Time of tree planting was set at the year zero. Based on a literature review by Paul et al. (2002), we included a lag phase of 5 years following tree planting. The accretion trajectories of tree-C presented within each Panel were projected using the unified first-order model developed in Fig. 6. The magnitude of tree-C gains as well as the turnover rates (k) and mean residence times (MRT) of prairie-C trajectories are provided within each panel. The Huron site (Panel A) had an adjacent native grassland available, which here is provided as a reference and plotted prior to the time of conversion into annual cropland. Note the different x and y scales across panels.