# Peer review of "subsequent afforestation of croplands"

_SOIL, 2021_

## Referee Comment (RC2)

**1. General comments**

In this study, **Nonlinear turnover rates of soil carbon following cultivation of native grasslands and subsequent afforestation of croplands**, Hernandez-Ramirez et al., use existing soil C data produced in the previous studies (Hernandez-Ramirez et al., 2011) in combination with new measurements of stable C isotopes to evaluate long-term C turnover rates in relation to land use changes, from grassland to cropland and subsequently from cropland to forest.

The paper reads well and I find this an interesting paper worth to be considered for publication in the SOIL journal.

The study is well introduced and the authors build on existing literature to highlight the effect of land use change on SOM. But I did not find information on previous related studies especially on long-term carbon turnover or approaches that have been used.

The authors did a good job on the materials and methods section. The study sites, the model, and the land use investigated are well described. However, the reasons behind the choice of the model and uncertainty related to the model are not clearly provided. Very little information is provided on basic soil physical-chemical properties that are known to influence C turnover such as texture, pH, oxides among others. These factors drive the C stabilization mechanisms which are briefly mentioned in the introduction and discussion. Having such information in the manuscript may highlight future studies as you mentioned in the discussion section.

The results section is clearly described. It has a lot of information that supports most of the statements made in the manuscript. The figures and tables are informative. Figure 4, the Y-axis is not scaled across panels. Is there a reason for this?

The discussion and conclusion are short but very informative. The authors did a good job here too. As for the materials, and methods, the authors mentioned the role of mineral and physical protection of soil C. Having information on the basic soil properties related to these mechanisms may strengthen some statements of the conclusions.

In the following sections, I will provide comments and suggestions for each section of the manuscript as indicated by lines.

**2. Specific comments**

Lines 83-84: You may consider revising this sentence. Does "biological-mediated decomposition" different from "decomposition"?

Line 86: Do "decomposition" and "mineralization" have different meanings in this context? One of them would suffice.

Lines 92, 94, 96, and 102 Authors mentioned "long-term" but it would be clear to the reader if this is clarified in terms of numbers.

Line: 112, Be precise by using numbers to reflect the chronosequence

Line 123: How deep was the core? Was it a one meter core or did the authors focus on 0-30cm? They also refer to previous studies for more details but it may help the reader if this information is added here.

Lines 143-145. It is good that excluding the contribution from erosion is supported with data on topography. But it would have been much better to provide this information here. The authors may consider saying "At our study sites, the slope gradient ranges from X to Y and the topography is classified as flat. Given the flat topography, we also assumed negligible C removals or additions due to erosion or deposition."

The authors assume that the C input and output are balanced. Are there references to support this statement? It might be from the previous studies in the region or similar conditions. Usually, long-term C turnover is accurately or well described with multiple-pools models. What makes Eq. [1] a suitable approach for this study? The authors may consider adding that information in the method section.

Lines 152-156: Excellent idea, to use the cross-validation method to assess the model performance. In this section, the authors may also provide information on the total number of observations in the dataset used in this process. Given that sample size can greatly affect a model.

Line 236: The authors assume negligible contribution to C storage by annual cropland. How small is the C input from these sources? Are they really negligible? Any data from previous studies that support this statement? Unless crop residues (straws) are taken from the field, they should contribute to some extent to the topsoil C.

Line 314: As a followup to the previous comment, it looks like there are contributions from the recently added plant residues. How did the authors separate the C isotopic signature of the original land use (native grassland) from the recent crop residues?

Line 323-324: Same as the previous comments related to the "contributions of crop residues recently added".

Line 456: I guess "study studies" should be "study sites".

Lines 509-511. This is a very interesting finding. Why would old prairie-C decompose faster compared to fresh C input from roots and litter? The remaining prairie-C should otherwise be considered as stable C (less labile)? This comment is also linked to the statement in lines 470-471. I did not see data on the C turnover rate of the forest input.

Lines 521-522: Sites located in Norfolk and Mead have comparable temperature and precipitation. Did the authors look at the mineralogical characteristics of these sites? The difference may also be related to the C stabilization mechanisms.

---

## Author Comment (AC1)

**Answers to review comments one - RC1**

soil-2021-5    Submitted on 16 Jan 2021

Nonlinear turnover rates of soil carbon following cultivation of native grasslands and subsequent afforestation of croplands

Guillermo Hernandez-Ramirez, Thomas J. Sauer, Yury G. Chendev, and Alexander N. Gennadiev

**Nonlinear turnover rates of soil carbon following cultivation of native grasslands and subsequent afforestation of croplands**

In this study, Hernandez-Ramirez et al. use data from a previous publication (Hernandez-Ramirez et al. 2011), coupled with new measurements of C stocks and stable C isotopes to model the trajectory of carbon losses and accruals resulting from grassland conversion to cropland and subsequent afforestation via shelterbelts, respectively, using first-order kinetics. Overall, I feel the manuscript is too long, and needs significant work to shorten it and improve its readability. I realize there is quite a lot going on in terms of explaining the different sites, methods undertaken, and results, but I still feel there is a lot of opportunity to cut text without losing any information. In addition, much of what is presented here has already been presented in earlier publications - though the degree of overlap is not entirely clear - so fewer words can be spent reiterating what has already been described in those publications to further aid in shortening the manuscript. Some more emphasis on what aspects of this work are novel or noteworthy, overall and compared to the previous publications on these sites, would strengthen it as well. What follows are section-by-section suggestions and comments.

Answer:

We are deeply grateful for the overall positive evaluation. We are glad to address the review comments and to implement changes for further improving the paper. We agree that the method and results section can be shortened in the revised manuscript by omitting information that was already presented in previous publications. For example, part of the site description of the

chronosequences and paired sites could be omitted by providing references as this information (e.g., plant species in lines L172) has been shown in previous publications.

Introduction:

Define "long-term" somewhere in the introduction.

Answer:

Certainly, we are glad to define this "term". For instance, in Lines 92 "(ranging from decadal to centurial scales)"

The introduction could use some background on SOM accrual rates, and the fact that accrual is assumed to be asymptotic, but that this is rarely tested/quantified. All of the popular models assume or produce this asymptotic behavior, but it's rarely been verified in studies of accrual that are longer than xx number of years.

Some background on any previous studies which have provided data on SOM accrual for longer-term periods (or lack thereof) would be useful here.

I think the trees/shelterbelts ideas could be more clearly linked to SOM accrual and the specific knowledge gap linking the two could be better identified.

Answer:

Yes, we are glad to include these ideas in the third paragraph of the introduction. "Testing SOM accrual rates over long-term (decades to centuries) is lacking in the literature. Studies such as Jastrow et al. (1996), Hernandez-Ramirez et al. (2011) and **Mary et al. (2020)** have previously evaluated soil C turnover rates as a function of changes in land management over one or a few decades; however, the underlying assumption of asymptotic behavior in the rate of soil C change has rarely been verified over longer periods such as centuries."

Mary, B., H. Clivot, N. Blaszczyk, J. Labreuche, F. Ferchaud. 2020. Soil carbon storage and mineralization rates are affected by carbon inputs rather than physical disturbance: evidence from a 47-year tillage experiment. Agric. Ecosyst. Environ., 299

Additionally, we consider that having afforestation effects on carbon sequestration and the dynamics of carbon accrual rates all together in one paragraph can become too much information to grasp at once, and hence, while aiming at readability, we decided to dedicate the second

paragraph of the introduction to such land use effects and the following paragraph (third paragraph) to the background and asymptotic behavior of soil carbon accrual rates over the long term.

Line 76: the phrase "soil microbiology is sustained" is a little awkward and I'd suggest rewording. Maybe "soil microbial communities are sustained and diversified". Or maybe something like "diverse microbial communities can flourish beneath mature trees"
Answer:

Thanks for the suggestion. We agree to include "diverse microbial communities can flourish beneath mature trees"

Lines 91-98 are a bit repetitive and don't convincingly explain why this knowledge gap needs to be filled. I think these lines could be compressed and the need for this knowledge could be more clearly/convincingly stated here. Maybe using logic similar to the above comment about background on SOM accrual.
Answer:

We agree that these repetitive statements need to be excluded. We are glad to use instead the following sentences:
"Moreover, the direction and net rates of SOM accrual as a response to land use changes need to be assessed in the long term (i.e., ranging from decadal to centurial scales) (Paustian et al., 1992; Hernandez-Ramirez et al., 2011). This new knowledge will inform how lasting these effects of land management options on soil C storage enable predictions of future soil C sequestration (Richter et al., 1999; Guo and Gifford, 2002). Our study endeavors to address and fill these knowledge gaps."

Methods:
There is no description of soil processing (it may be in the previous publications cited, but I think  it still should be stated explicitly here). Were the samples sieved, and roots and plant fragments removed prior to analysis? Was any organic material or O horizon removed?
Answer:

We are glad to explicitly include the soil processing.

"Field moist soil samples were passed through 8- and 2-mm sieves, air dried, and ground with a roller mill (Bailey Manufacturing Inc., Norwalk, IA) to create a fine powder consistency. Identifiable plant materials were removed prior to grinding (Hernandez-Ramirez et al., 2011; Chendev et al., 2015b)."

In the case of the forest floor, this carbon is not considered as permanent C storage (but part of the flow of carbon from plants into the soil), and hence, we herein focused on actual soil carbon. Earlier work published by Sauer et al. (2007) in the Mead site in Nebraska provided comprehensive quantification of forest-floor carbon in the shelterbelt.

Line 160: replace "studies" with "studied"
Answer:
Certainly – thanks.

Lines 181-186: I like that multiple scenarios were evaluated, but some explanation of why these two scenarios were chosen would be helpful (why 95% and not some other %age?)
Answer:
Good point. We are glad to capture this input by adding the following sentences in the text of the method description: "The rationale for implementing the 95% scenario arises from the uncertainty of whether full C equilibrium has been reached. As the final phases in an asymptotic trajectory are incremental, we undertook the evaluation of a narrow but identifiable deviation below full C equilibrium. The choice of 95% represents that an analytical precision for quantifying soil carbon can typically be found within 5%. In other words, based on a principle of detection limit in soil C measurements, a 95% can be considered a minimal deviation from full equilibrium (100%) that is already discernable, but still related to the general variability of the quantification method and associated results. Hence, we took the freedom to assess this plausible range at and below full theoretical equilibrium, with the soil C storage having reached 100% or 95% of the ceiling capacity, respectively. From a broader perspective, this alternative 95% scenario also explores and represents the prospect that the soils beneath the shelterbelt could still be incrementally accruing C even 55 years after tree planting."

Line 198: Change "tall prairie" to "tallgrass prairie"

Answer

Yes, we agree.

Lines 213: Replace "Normal" with "Mean" if you are presenting mean annual precipitation.

Answer

Yes, we agree. And we are further clarifying that these are long term "means".

Line 214: Replace "Normal air temperature" with "Mean annual air temperature" if that is what you are presenting.

Answer

Yes, we agree. And we are further clarifying that these are long term "means".

Line 236: "...assuming that net contributions to soil C storage were negligible…" gives me pause. To clarify, you are assuming that there is no total increase in C from crop-derived C inputs, and that allows for these quantification approaches to be used. But there will certainly be contributions of crop-derived C to the C pools, even if they are not increasing in size (see xxx). Can you clarify whether you are assuming no contributions at all (i.e. no change in ð□›…13C from that of the native systems during cropping) or you are allowing for that change in ð□›…13C but assuming no net increase in total C? This relates to the following comment: Lines 250-258: does "ð□›…13C native soil" refer only to the soil under native vegetation, or just whatever vegetation was present before afforestation? I'm finding this wording confusing because the cropped soils are also being used to represent the "ð□›…13C native soil" here (lines 251-252). I think the wording here needs to be clarified or better explained, since this is key to the interpretation of the results. Based on line 395 and the caption of Fig. 5, it seems as though the authors are assuming "the rest of the soil C was attributed to remaining prairie C", but this can not be the case if crop inputs contributed to the C there, which they surely will have.

Answer

We are glad to rephrase and clarify Line 235-228 as: "Based on this legacy contribution of prairie vegetation to soil C over the Holocene, approaches based on stable isotope signatures became feasible in the three sites in United States. Furthermore, the ability to use a C stable

isotope approach to partition the current soil C into two specific C pools (i.e., remaining prairie-C and new tree-C) requires a constraining assumption that much of the plant residues added yearly over the annual cropping stages decomposes during the following growing season (Gregorich et al., 2017). Therefore, this premise entails that most of C in crop residues enters the soil to become lost back to the atmosphere within a year, and hence, having near-negligible contributions to changes in both net C accrual and $\delta^{13}C$ isotopic composition in the soil."
Gregorich, E.G., H. Janzen, B.H. Ellert, B.L. Helgason, B. Qian, B.J. Zebarth, D.A. Angers, R.P. Beyaert, C.F. Drury, S.D. Duguid, W.E. May. 2017. Litter decay controlled by temperature, not soil properties, affecting future soil carbon. Glob. Chang. Biol., 23:1725-1734

It is re-emphasized that this assumption is necessary to implement the kinetics modelling in this study while allocating soil C into two pools. Furthermore, the annual croplands adjacent to the evaluated afforested areas were managed with tillage operations. During long-term recurrent tillage plowing and cultivation (as shown in Fig. 3 over 250 years), the amount of carbon loss from the topsoil largely exceeds the amount of new organic carbon derived from any contribution or transformation of plant residues into humus. As a result, a long-term trend of carbon losses was manifested.

As noted above, the underlying premise is that although every year there is a carbon input from crop residue, this carbon from crop residue is largely lost to the atmosphere with no significant net contribution to the total carbon on an annual basis. Hence, the crop residues do not influence the total carbon in the long term. Although not fully conclusive, good evidence for supporting this premise is in figure 3 for the Russian sites (and figure 2): the total soil carbon declines over the long term (decades and centuries) in the cropland sites; this is attributed to the fact that crop residue has minimal contribution to the total soil carbon and also because disturbance by tillage is increasing soil organic matter decomposition and release to the atmosphere. Moreover, isotope analyses in the US sites suggested that nearly all of the carbon in the croplands resembles the carbon of the grasslands instead indicating an influence of crop residues, while the tree-C contributions really shifted the isotopic composition to enable identification of carbon sources (e.g., Huron). Several references support this collective premise (Jastrow et al., 1996; Follett et al., 1997; Hernandez-Ramirez et al., 2011; Gregorich et al., 2017; Mary et al., 2020). In addition, this implicit assumption is actually required to enable kinetic modelling to allocate total carbon

into two carbon pools (tree-C vs. remaining prairie-C). In other words, C added annually by crop residues was assumed to be decomposed and released by the end of every agricultural year; therefore, although crop residues are adding carbon to the soil at some point during the year, this added-C is gradually lost over the following season (with minimal or even without any net carbon retention from crop residue in the long-term).

From another perspective, the kinetics modelling approach used in the study (equation [2]) enables allocating two specific carbon pools (e.g., remaining prairie-C vs. tree-C). In other words, the underlying assumption is that the crop residue carbon that entered the system every year undergoes decomposition within a season leaving no net contribution to the total carbon in the long term (Gregorich et al., 2017). Crop residue in these croplands are not actually increasing soil carbon content over time; the soil carbon in these annual croplands is actually gradually decreasing (figure 3 and figure 2). To separate and quantify any potential contribution of crop residue to total carbon would have required an additional set of different measurements at our sites; for example, isotopically-labelling plant residue and tracking its decomposition or retention over multiple years.

As noted above, this (and others) assumption are required for implementing and constraining the kinetics modelling as in equation [1] (Jastrow et al., 1996; Follett et al., 1997; Hernandez-Ramirez et al., 2011).

Results:
In general I found this section to be long and difficult to follow at times. I recommend the authors condense much of the writing and work to make it easier to digest.
Line 288: Remove "(A)."
Answer:
Agree. Thanks for catching this.

Line 290: Change "(Fig. 2A, Fig. 2B, and Fig. 2C)" to (Fig. 2A-C), here and elsewhere (e.g., line 300).
Answer:
Agree

Line 296: Can you give the time spans here for reference?

Answer:

Agree. We are glad to include "the three chronosequences (up to 250 years), net soil"

Line 307: change "which can question" to "calling into question the" or "challenging the"

Answer:

Agree. "challenging the"

Lines 311-315: Sentence beginning on line 311 is not a full sentence. Maybe "while" need to be removed? Or this needs to be combined with the following sentence. In general these two sentences are a bit awkwardly placed and might make more sense if placed elsewhere.

Answer:

Yes, we agree to remove "while" from line 311.

Line 323-324: I think it would be good to clarify that this is only in the soil underneath the trees, rather than xx% C accrual for the whole field.

Answer:

Yes, we agree. We are glad to expand this explanation as follows: "…, afforestation replenished on average 81% of these cropping-induced C losses, specifically in the soils beneath the trees…"

Line 318-328: If this data (Table 1) is the subject of another paper, it might be better to summarize it only very briefly and simply refer to the table.

Answer:

Yes, we agree. The revised manuscript will include:

"

Based on the assessed pairwise comparisons, afforestation in the form of shelterbelts replenished soil C storage after long-term annual cropping had led to decreasing soil C compared with adjacent native grasslands (Table 1). In further details, of the substantial soil C storage that had been depleted over time during annual cropping (i.e., -18.9 ± 5.3 Mg C ha$^{-1}$), afforestation replenished on average 81% of these cropping-induced C losses (Table 1).

"

These changes reduced the word count from 159 words to now only 70 words.

Answer:

This normalization aims at having all the data between 0 and 1. We are glad to add in line 332: "After normalizing all cropland-chronosequence data (i.e., zero to one; dimensionless), turnover…". Normalization here implies bringing the data shown in figures 2A-C into the very same scale by dividing the carbon storage density in the cropland locations by the carbon storage in the native grassland within each of the 3 chronosequences separately, and then (after normalizing each dataset between 0 and 1), we pulled the data of the 3 chronosequences together for a collective analysis. This process enabled robust exploration of the carbon trajectory over 250 years (in Figure 3). This normalization step brings all data to the same scale within the range from zero to a maximum of 1.0.

Answer:

This is the same normalization explained in the previous answer and also shown in Figure 3, and hence these values 1-0.711 are unitless.

Line 350-353: This sentence is confusing to me. It is not clear what the 81% and the 94.5% represent - are they different timepoints or maybe different sites? After multiple readings I think I understand this to mean that after replenishing 81% of the C lost during cropping, the shelter belts had 94.5% of the C of the native grassland. I think you can word this more clearly.

Answer:

Thanks for this valuable suggestion. We are glad to reword this passage in the revised manuscript.

"After replenishing 81% of the C lost during long-term annual cropping, the shelterbelts had 94.5% of the initial C of the native grassland (i.e., $0.81 \times 0.289 + 0.711 = 0.945$; Table 1)."

Answer:

We are glad to clarify this statement as follows:

"…, the stable isotope approach followed by mass balance effectively allocated and quantified the tree-derived soil C,…"

Line 421: "stoichiometry of absolute capacities" is an odd phrase and I am not sure I understand what it means. "Proportionality of the initial prairie-C" is also a little odd, so I would suggest rewriting this sentence to be a little easier to understand.

Answer:

We are glad to update this text to improve understandability and readability as follows:

"This can indicate that the net changes in whole soil C storage under afforestation: i) simultaneously encompassing the noted prairie-C losses and the asymmetrically-larger tree-C gains, and ii) and did not follow a fixed proportionality to the initial prairie-C."

Ind 456: remove "study"

Answer:

We agree. Thanks!

Table 1. Perhaps add columns that shows the amount of time for each land use, or say in the caption?

Answer:

We agree to add in the table title as follows: "At time of soil sample collection, the ages of the annual croplands in Streletskaya and Yamskaya were at least 140 years, and at least 145 years of age in Kamennaya. In all three sites, the shelterbelts had been planted 55 years prior to soil sample collection."

Many of the figure captions are quite long. I think they can be shortened significantly without losing any key information.

Answer:

We agree to revise the figure captions for shortening them. Yes, we agree to remove several methodological aspects from the various figure captions as they are represented already within the text of the method description sections.

Fig. 3. The Y axis wording is a little odd, maybe "Proportion of original C" or similar?
Answer:
We agree to update this Y axis title to: "Proportion of Original C in the Native Grasslands".

Discussion:
Overall, this section feels a little disjointed and lacks a good flow of ideas from paragraph to paragraph. Restructuring it so that it has a clearer arc and/or follows a similar flow of ideas as the introduction would help make the ideas easier to digest and follow.
Lines 492-497: This is a pretty clear explanation and I think it would be useful to have this much earlier, in the methods. I found it really hard to figure out whether this was the case from reading the methods section.
Answer:
Certainly, we are glad to move this clear explanation to the method section (towards the end of the method section around line L 276).

Lines 516-517: Rather than saying "trees generated the conservation of the entire prairie- legacy" I would say something like "afforestation did not lead to loss of prairie-C" or "prairie-C was conserved during afforestation".
Answer:
We agree to update this sentence; we will be using "In the case of Huron site, afforestation conserved the initial prairie-C while also contributing directly to additional tree-C accrued in an overall increasing SOM pool (Fig. 7A)"

Line 527: I think it would make more sense to use "climatic" instead of "hydrothermal", and do this throughout the discussion. Could soil texture or aggregation play a role here?
Answer:
We agree to use climatic (instead of hydrothermal) both in L527 and L557

We have now included a new Table including soil texture data.

The following is the new table

Table 1. Topsoil textures and pH in the 9 study sites. Numerals correspond to sites as shown from west to east in Fig. 1.

| Site | Soil pH | Soil texture |
|---|---|---|
| 1. Huron† | 7.0 | sandy loam |
| 2. Norfolk† | 6.8 | loamy sand |
| 3. Mead† | 6.1 | silty clay loam |
| 4. Streletskaya Steppe‡ | 7.0 | loam |
| 5. Ivnyanskiy‡ | 7.5 | silt loam |
| 6. Prokhorovskiy‡ | 7.2 | loam |
| 7. Gubkinskiy‡ | 7.4 | clay loam |
| 8. Yamskaya Steppe‡ | 7.2 | loam |
| 9. Kamennaya Steppe‡ | 7.6 | clay loam |

† correspond to the cropland location within this site.
‡ correspond to the native grassland location within this site.

We agree with the suggestion to include additional soil property data. We do have soil texture and pH data available, and we are indeed glad to include the available data in the revised paper (please see above the new table 1 – to be included in the revised manuscript). We note that we have already included texture and pH information for the US sites in the method section of the original submission at L215-217. Although we agree that existing literature supports that soil properties drives soil carbon pools and turnover (as shown for instance in Quesada et al. 2020). We note that in the specific cases of our study sites, carbon turnover data does not relate well with available soil texture and pH data. This is likely because the reduce sample size in our study sites (e.g., the 3 Russian chronosequence sites shown in Figure 2 is contrasting to the 14 soil classes sampled by Quesada et al. 2020). Having said this, we will include available soil texture and pH data in our revised manuscript to better characterize all 9 sites including Russia and US.

Quesada, C. A., Paz, C., Oblitas Mendoza, E., Phillips, O. L., Saiz, G., and Lloyd, J. 2020. Variations in soil chemical and physical properties explain basin-wide Amazon forest soil carbon concentrations, SOIL, 6, 53–88

Lines 530-536: What about physicohemical protection mechanisms? SOM fractionation might give additional insights, and I'd suggest that would be another important avenue for future research.

Answer:

We completely agree that "SOM fractionation approaches offer excellent avenue for further unravelling the stabilization mechanisms of C in the soil". We are glad to highlight this at this same location Lines 536 as this will entice future research on the subject.

Line 540-541: Change "shown to continually being lost" to "was shown to be lost continually"

Answer:

We agree.

Line 541-542: This sentence seems to contradict the prior and following sentences. This makes it sound like not much of the prairie C was lost (only 1.7% of the original prairie C over 55 years is a very small amount). How does this compare to the rate of loss of native prairie C in the cropland? Is it really something to note, or is it a relatively normal turnover rate for old C?

Answer:

We agree to further elaborate on this aspect to clarify this explanation. We are glad to clarify these sentences as follows:

"

Our turnover estimations using kinetics modelling suggested that only 1.7% of the initial grassland-C was lost over these 55 years following shelterbelt afforestation (Fig. 3). This relatively small grassland-C loss is in part because soil C had been depleted over nearly two centuries of annual cropping prior to tree planting. Nevertheless, mycorrhizae activity in afforested soils can preferentially access and utilize remaining grassland-C beneath trees (Mellor et al., 2013). Hence, this biological effect could contribute to gradual decreases in remaining grassland-C in afforested soils.

"

Lines 570-574: This is a nice explanation and something to this effect should be included in the introduction (either move this text there, or introduce the idea there and reiterate it here).

Answer:

We agree. We are glad to include this nice explanation as a new sentence located towards the middle of the third paragraph in the introduction section:

"

Nevertheless, since using linear rates to describe long-term changes in soil C can often misestimate C inventories and sequestration, nonlinear rates of soil C accrual need to be derived to accurately predict the trajectory of soil C changes with time following land use conversions (Post and Kwon, 2000; Garten 2002).

"

Lines 587, 591, 592: I have only ever seen "MRT" refer to the C itself, never an MRT of soil C accretion or depletion. What does this mean? I think it would make more sense to refer to the rate of accretion/depletion, or average time to xx% of total accretion or something like that.

Answer:

We certainly agree to clear up this confusion. We are now using similar wording at these various locations as one of these: MRT or "rate of accretion" or "rate of depletion" as appropriate at each location (instead of the confusing expression: "MRT of soil C accretion or depletion").

Paragraph beginning on line 604: I agree with the suggestion of studying different physical pools of C in terms of stability, and I like the ideas introduced here. However, they seem to come out of nowhere given that they have not been introduced earlier in the manuscript. I might suggest including them earlier on in the discussion (e.g. when discussing potential differences in native prairie C turnover between sites) and then coming back to it here to expand on it.

Answer:

We agree. We are now emphasizing the need to study different C pools and their stability through SOM fractionation earlier in the discussion section 4.1 - fourth paragraph near Line 536.

Summary:

The authors state that afforestation is a suitable means to address climate change, but it would be helpful to put their results in context of the amount of area being afforested in these cases. While there was significant C accrual in the soils under the trees, the area being planted to trees seems relatively small, so what is the overall gain from this practice? Is it really very significant given that most of the cropland is not being planted to trees? I think that noting these relative area sizes and total C losses and gains in the croplands vs afforested areas is very important to put the results into larger context.

Answer:

We agree that informing typical proportion of afforested vs. cropped area provides good context for the readers. We are adding the following sentences in the middle of the summary paragraph:

"

While our study confirmed these substantial C accruals in the soils under the trees, the overall gain at the actual landscape scale will depend in part on the proportion of farmland dedicated to afforestation, with afforested areas typically accounting for up to 5% of the farmlands (Amadi et al., 2016).

"

In line with this comment, we also propose including in the text of the summary the following sentences:

"Our focus on soil organic C behavior in soils under shelterbelts is only part of a broader range of studies evaluating the overarching impacts of agroforestry on soil quality and crop yield across the landscape. Beyond C sequestration, the benefits of shelterbelts can also be manifested in improving the local climate as well as increasing crop yields."

---

## Author Comment (AC2)

**Answers to review comments one - RC2**

soil-2021-5      Submitted on 16 Jan 2021

Nonlinear turnover rates of soil carbon following cultivation of native grasslands and subsequent afforestation of croplands

Guillermo Hernandez-Ramirez, Thomas J. Sauer, Yury G. Chendev, and Alexander N. Gennadiev

General comments

In this study, Nonlinear turnover rates of soil carbon following cultivation of native grasslands and subsequent afforestation of croplands, Hernandez-Ramirez et al., use existing soil C data produced in the previous studies (Hernandez-Ramirez et al., 2011) in combination with new measurements of stable C isotopes to evaluate long-term C turnover rates in relation to land use changes, from grassland to cropland and subsequently from cropland to forest.

The paper reads well and I find this an interesting paper worth to be considered for publication in the SOIL journal.

The study is well introduced and the authors build on existing literature to highlight the effect of land use change on SOM. But I did not find information on previous related studies especially on long-term carbon turnover or approaches that have been used.

Answer:

We are deeply grateful for the overall positive evaluation. We certainly agree to include additional information from earlier studies focusing on long-term carbon turnover in the introduction section (within the third paragraph of the introduction). These will include Huggins et al. (1998) and Collins et al. (1999).

Collins, H.P., R.L. Blevins, L.G. Bundy, D.R. Christenson, W.A. Dick, D.R. Huggins, and E.A. Paul. 1999. Soil carbon dynamics in corn-based agroecosystems: Results from carbon-13 natural abundance. Soil Sci. Soc. Am. J. 63:584–591.

Huggins, D.R., C.E. Clapp, R.R. Allmaras, J.A. Lamb, and M.F. Layese. 1998. Carbon dynamics in corn–soybean sequences as estimated from natural carbon-13 abundance. Soil Sci. Soc. Am. J. 62:195–203.

The authors did a good job on the materials and methods section. The study sites, the model, and the land use investigated are well described. However, the reasons behind the choice of the model and uncertainty related to the model are not clearly provided. Very little information is provided on basic soil physical-chemical properties that are known to influence C turnover such as texture, pH, oxides among others. These factors drive the C stabilization mechanisms which are briefly mentioned in the introduction and discussion. Having such information in the manuscript may highlight future studies as you mentioned in the discussion section.

Answer:

Thanks for the positive suggestions. The reasons for implementing the choice of kinetics model include the reduced numbers of model parameters and the capacity to constrain the upper and lower limits of the carbon trajectories (Jastrow et al., 1996); these collectively led to enhanced fitting and the ability to successfully conduct cross-validation (as shown in Figure 2D). We are glad to include these concepts in the method section of the revised ms.

Jastrow, J.D. 1996. Soil aggregate formation and the accrual of particulate and mineral-associated organic matter. Soil Biol. Biochem, 28:665-676

We also agree with the comment suggesting to include additional soil property data. We do have soil texture and pH data available, and we are indeed glad to include the available data in the revised paper (please see below the new table 1 – to be included in the revised manuscript). We note that we have already included texture and pH information for the US sites in the method section of the original submission at the location L215-217. Although we agree that existing literature supports that soil properties drive soil carbon pools and turnover (as shown for instance in Quesada et al. 2020), we note that in the specific cases of our study sites carbon turnover data does not correlate with available soil texture and pH data. This is likely because the reduce sample size in our study (e.g., the 3 Russian chronosequence sites shown in Figure 2 is contrasting to the 14 soil classes sampled by Quesada et al. 2020). Having said this, we will

include available soil texture and pH data in our revised manuscript to better characterize all 9 sites including Russia and US.

Quesada, C. A., Paz, C., Oblitas Mendoza, E., Phillips, O. L., Saiz, G., and Lloyd, J. 2020. Variations in soil chemical and physical properties explain basin-wide Amazon forest soil carbon concentrations, SOIL, 6, 53–88

The following is the new table

Table 1. Topsoil textures and pH in the 9 study sites. Numerals correspond to sites as shown from west to east in Fig. 1.

| Site | Soil pH | Soil texture |
| --- | --- | --- |
| 1. Huron† | 7.0 | sandy loam |
| 2. Norfolk† | 6.8 | loamy sand |
| 3. Mead† | 6.1 | silty clay loam |
| 4. Streletskaya Steppe‡ | 7.0 | loam |
| 5. Ivnyanskiy‡ | 7.5 | silt loam |
| 6. Prokhorovskiy‡ | 7.2 | loam |
| 7. Gubkinskiy‡ | 7.4 | clay loam |
| 8. Yamskaya Steppe‡ | 7.2 | loam |
| 9. Kamennaya Steppe‡ | 7.6 | clay loam |

† correspond to the cropland location within this site.
‡ correspond to the native grassland location within this site.

The results section is clearly described. It has a lot of information that supports most of the statements made in the manuscript. The figures and tables are informative. Figure 4, the Y-axis is not scaled across panels. Is there a reason for this?

Answer:

Thanks for noting this aspect of the Y-axis. We have made the y-scales across panels to be different to be able to zoom in and to show more details to the readers. We are going ahead with noting this aspect in the figure caption.

The discussion and conclusion are short but very informative. The authors did a good job here too. As for the materials, and methods, the authors mentioned the role of mineral and physical protection of soil  C. Having information on the basic soil properties related to these mechanisms may strengthen some statements of the conclusions.

Answer:

As noted in an answer above, we agree to include available data of soil texture and pH in the body of the revised manuscript as per new Table 1 above (although there were no clear patterns between carbon turnover and these properties; likely due to the sample size as mentioned above).

In the following sections, I will provide comments and suggestions for each section of the manuscript as indicated by lines.

Specific comments

Lines 83-84:  You may consider revising this sentence. Does "biological-mediated decomposition" different from "decomposition"?

Answer:

We agree with keeping only "decomposition" and removing "biological-mediated" in this sentence.

Line 86: Do "decomposition" and "mineralization" have different meanings in this context? One of them would suffice.

Answer:

We agree with keeping only "mineralization" and removing "decomposition" in this sentence.

Lines 92, 94, 96, and 102 Authors mentioned "long-term" but it would be clear to the reader if this is clarified in terms of numbers.

Answer:

This is a good clarification. We are glad to include in the text of this paragraph in the introduction section "(i.e., ranging from decadal to century scales)".

Line: 112, Be precise by using numbers to reflect the chronosequence

Answer:

Yes, we agree. We are switching to use numerals (instead of words) to express the number of chronosequences. We are doing this at this location in the text and elsewhere.

Line 123: How deep was the core? Was it a one meter core or did the authors focus on 0-30cm? They also refer to previous studies for more details but it may help the reader if this information is added here.

Answer:

The soil samples were collected to at least "1 meter depth". We are glad to include this info in the method section of the revised manuscript.

Lines 143-145. It is good that excluding the contribution from erosion is supported with data on topography. But it would have been much better to provide this information here. The authors may consider saying "At our study sites, the slope gradient ranges from X to Y and the topography is classified as flat. Given the flat topography, we also assumed negligible C removals or additions due to erosion or deposition."

Answer:

We agree and we are glad to include the following sentences. "At the various study sites, the terrain slopes ranged up to 2%, with the exception of Huron site that had a 3% slope. Hence, the general topography in our study sites was classified as flat. As most sites are considered semi arid, water erosion is assumed minimal; likewise, enough vegetative cover limits wind erosion. Given the dominant flat topography and low rainfall amounts, we also assumed negligible C removals or additions due to erosion or deposition.

The authors assume that the C input and output are balanced. Are there references to support this statement? It might be from the previous studies in the region or similar conditions. Usually, long-term C turnover is accurately or well described with multiple-pools models. What makes Eq. [1] a suitable approach for this study? The authors may consider adding that information in the method section.

Answer:

The assumptions of (i) C input = C output and (ii) steady state conditions (i.e., $\delta C / \delta t = 0$) are required to implement the kinetics modelling in equation [1] (Jastrow et al., 1996; Follett et al., 1997; Hernandez-Ramirez et al., 2011). As suggested, we are glad to clarify these references and information in the text of the method section. Furthermore, the kinetics modelling and stable isotope approaches used in the study enable distinguishing two soil carbon pools; for instance, for soil carbon changes under afforestation, total carbon can be partitioning between previously-existing carbon (e.g., remaining prairie-C) and newly-accrued carbon (e.g., new tree-C). We agree that carbon turnover is more accurately quantified in the long-term compared with short-term carbon turnover measurements.

Lines 152-156: Excellent idea, to use the cross-validation method to assess the model performance. In this section, the authors may also provide information on the total number of observations in the dataset used in this process. Given that sample size can greatly affect a model.

Answer:

Certainly, we are glad to include the sample size in this test (total number of observations in the dataset). This is "(n= 6)" at the end of the sentence in location L155.

Line 236: The authors assume negligible contribution to C storage by annual cropland. How small is the C input from these sources? Are they really negligible? Any data from previous studies that support this statement? Unless crop residues (straws) are taken from the field, they should contribute to some extent to the topsoil C.

Answer:

As noted above, the underlying premise is that although every year there is a carbon input from crop residue, this carbon from crop residue is largely lost to the atmosphere with no net contribution from crop residue to the total carbon on an annual basis. Hence, because of annual decomposition, the crop residues do not influence the total carbon in the long term. Although not fully conclusive, good evidence for supporting this premise is in figure 3 for the Russian sites (and figure 2): the total soil carbon declines over the long term (decades and centuries) in the cropland sites; this is attributed to the fact that crop residue has null or minimal contribution to the total soil carbon and also because disturbance by tillage is increasing soil organic matter decomposition and release to the atmosphere. Isotope analyses in the US sites showed that nearly all the carbon in the croplands resembles the carbon of the grasslands instead of an influence of crop residues, while the tree-C contributions really shifted the isotopic composition to enable identification of carbon sources (e.g., Huron). Several references support this collective premise (Jastrow et al., 1996; Follett et al., 1997; Hernandez-Ramirez et al., 2011; Gregorich et al., 2017; Mary et al., 2020). In addition, this implicit assumption is actually required to enable kinetics modelling to allocate total carbon into two carbon pools (tree-C vs. remaining prairie-C). In other words, C added annually by crop residues was assumed to be decomposed and released by the end of every agricultural year; therefore, although crop residues are adding carbon to the soil at some point during the year, this added-C is gradually lost over the following season (with minimal or even without any net carbon retention from crop residue in the long-term). We agree to capture and include these various concepts in the text of the revised manuscript.

Gregorich, E.G., H. Janzen, B.H. Ellert, B.L. Helgason, B. Qian, B.J. Zebarth, D.A. Angers, R.P. Beyaert, C.F. Drury, S.D. Duguid, W.E. May. 2017. Litter decay controlled by temperature, not soil properties, affecting future soil carbon. Glob. Chang. Biol., 23:1725-1734

Mary, B., H. Clivot, N. Blaszczyk, J. Labreuche, F. Ferchaud. 2020. Soil carbon storage and mineralization rates are affected by carbon inputs rather than physical disturbance: evidence from a 47-year tillage experiment. Agric. Ecosyst. Environ., 299

It further is re-emphasized that this assumption is necessary to implement the kinetics modelling in this study while allocating soil C into two pools (tree-C vs. remaining prairie-C). Moreover, the annual croplands adjacent to the evaluated afforested areas were managed with tillage operations. During long-term recurrent tillage plowing and cultivation (as shown in Fig. 3 over 250 years), the amount of carbon loss from the topsoil largely exceeds the amount of new organic carbon derived from any contribution or transformation of plant residues into humus. As a result, a long-term trend of carbon losses was manifested.

Line 314: As a followup to the previous comment, it looks like there are contributions from the recently added plant residues. How did the authors separate the C isotopic signature of the original land use (native grassland) from the recent crop residues?

Answer:

As noted in the answer immediately above, the kinetics modelling approach used in the study (equation [2]) enables allocating two specific carbon pools (e.g., remaining prairie-C vs. tree-C). In other words, the underlying assumption is that the crop residue carbon that entered the system every year undergoes decomposition within a season leaving no significant net contribution to the total carbon in the long term (Gregorich et al., 2017). Crop residue in these croplands are not actually increasing soil carbon content over time; the soil carbon in these annual croplands is actually gradually decreasing (figure 3 and figure 2). To separate and quantify any potential contribution of crop residue to total carbon would have required an additional set of different measurements at our sites; for example, isotopically-labelling plant residue and tracking its decomposition or retention over multiple years.

Line 323-324: Same as the previous comments related to the "contributions of crop residues recently added".

Answer:

In Line 323-324, we are simply re-stating that growing trees increase soil carbon as opposed to annual cropland (which actually generated major carbon decreases). Some of the concepts in the previous answers above further address this comment.

Line 456: I guess "study studies" should be "study sites".

Answer:

Thanks for pointing this out. Yes, we are glad to correct this in the text of the revised manuscript.

Lines 509-511. This is a very interesting finding. Why would old prairie-C decompose faster compared to fresh C input from roots and litter? The remaining prairie-C should otherwise be considered as stable C (less labile)? This comment is also linked to the statement in lines 470-471. I did not see data on the C turnover rate of the forest input.

Answer:

Yes, we agree that this is a very fascinating finding. We can now see that there was a confusion in the original text, and we would like to clarify in the revised text that additions of fresh plant-C decompose faster than old prairie-C in the soil. At lines 509-511, we are glad to revise the text as follows: "This analysis showed that under afforestation, soil C remaining from original native grasslands continues to be lost from the profile likely via microbial mineralization (Fig. 3, Fig. 7). It is noted that the accretion and turnover of recently-added tree-C is much faster than these observed losses of remaining prairie-C beneath trees as the recently-added plant-C is considered relatively more labile than prairie-C."

In the case of the C turnover rate of the tree-C in afforested soils, this key result is explained in the paragraph located in lines from L455 to L475 as well as the C turnover rate is displayed in the Fig. 6.

Lines 521-522: Sites located in Norfolk and Mead have comparable temperature and precipitation. Did the authors look at the mineralogical characteristics of these sites? The difference may also be related to the C stabilization mechanisms.

Answer:

Norfolk and Mead are geographically close to each other (Fig. 1), and hence their general climates are similar. As noted in lines L213 - L215, precipitation in Norfolk and Mead were 696 and 747 mm yr-1, and temperature were 9.6, 9.9 °C, respectively. Although they are similar, Norfolk is slightly cooler and drier. In conjunction with Huron (which is even cooler and even drier), we noted this general regional pattern in lines L521 - L522. Based on first principles, we agree that soil properties are expected to drive the carbon accrual and mechanisms for carbon stabilization in the long term. As noted in the new Table 1, the texture of Norfolk is extremely sandy while Mead had a fine-textured soil; therefore, it seems that in the case of the relatively small sample size in this study (n=3; Huron, Norfolk and Mead), carbon accrual did not relate to soil properties, but instead there is a clear relationship between 'age since tree planting' and 'carbon gains in the soil profile' where the oldest site had the highest carbon accrual (Norfolk), the youngest site had the lesser carbon accrual (Huron), and Mead with an intermediate carbon accrual since it was intermediate in age of tree planting. The tree-C input over time can be so large and the modification of both microclimate and reduced disturbance so pronounced that these tree-related factors can be overriding other putative effects of soil properties on soil carbon storage, turnover, and stabilization.

---

## Author Response (AR2)

**Answer to Editor comments**

soil-2021-5     Submitted on 16 Jan 2021

Nonlinear turnover rates of soil carbon following cultivation of native grasslands and subsequent afforestation of croplands

Guillermo Hernandez-Ramirez, Thomas J. Sauer, Yury G. Chendev, and Alexander N. Gennadiev

*Executive Editor Decision: Publish subject to technical corrections (18 Jun 2021) by Jeanette Whitaker*

*Comments to the Author: Dear Authors*

*Many thanks for submitting this paper to SOIL. I am pleased to inform you that after the revision process your manuscript is now accepted for publication in SOIL, there is just one minor revision point requested by the topical editor which requires attention.*

*I look forward to seeing the paper published.*

*with best wishes*

*Jeanette Whitaker*

*Executive Editor Topical Editor Decision: Publish subject to technical corrections (16 Jun 2021) by Carolina Boix-Fayos*

*Comments to the Author:*

*Dear Authors,*

*Many thanks for addressing all the changes so efficiently. I think you have made a very good job. I just have a very minor question on your statement on line 162, and I would like if you could consider rewording the sentence, please.*

*My doubt is that it is highly debatable that the semi-arid factor is the cause of low erosion, it is possible that this is one of the reasons in your study areas, but in many other semiarid areas, there are torrential rainfalls concentrated in some moments of the year, leading to high rates of erosion. Thus I suggest changing the phrases in lines 162 and 163 as follows (or in a similar way):*

*"We assume that semiarid climate, enough vegetation cover and low slope limit water and wind erosion"*

*Furthermore I suggest to change the last section called "Summary" into "Conclusions", I think the style of the paragraphs respond more to a conclusion with a "take-home message" than to a summary.*

*Cheers, Carolina*

**Answers to Editor comments:**

Thanks for the valuable feedback. We sincerely appreciate the acceptance.

We agree and have now implemented these two changes in the updated manuscript version. We included "We assume that semiarid climate, enough vegetation cover and low slope limit water and wind erosion" as well as "Conclusions".